# AVHBench: A Cross-Modal Hallucination Benchmark for Audio-Visual Large Language Models

**Kim Sung-Bin**[1*]  **Oh Hyun-Bin**[1*]  **Lee Jung-Mok**[1]  **Arda Senocak**[2]
**Joon Son Chung**[2]  **Tae-Hyun Oh**[1,3,4]

[1]Dept. of Electrical Engineering and [3]Grad. School of Artificial Intelligence, POSTECH
[2]School of Electrical Engineering, KAIST  [4]School of Computing, KAIST

## Abstract

Following the success of Large Language Models (LLMs), expanding their boundaries to new modalities represents a significant paradigm shift in multimodal understanding. Human perception is inherently multimodal, relying not only on text but also on auditory and visual cues for a complete understanding of the world. In recognition of this fact, audio-visual LLMs have recently emerged. Despite promising developments, the lack of dedicated benchmarks poses challenges for understanding and evaluating models. In this work, we show that audio-visual LLMs struggle to discern subtle relationships between audio and visual signals, leading to *hallucinations* and highlighting the need for reliable benchmarks. To address this, we introduce `AVHBench`, the first comprehensive benchmark specifically designed to evaluate the perception and comprehension capabilities of audio-visual LLMs. Our benchmark includes tests for assessing hallucinations, as well as the cross-modal matching and reasoning abilities of these models. Our results reveal that most existing audio-visual LLMs struggle with *hallucinations caused by cross-interactions between modalities*, due to their limited capacity to perceive complex multimodal signals and their relationships. Additionally, we demonstrate that simple training with our `AVHBench` improves robustness of audio-visual LLMs against hallucinations. Dataset: https://github.com/kaist-ami/AVHBench

## 1  Introduction

We live in the world filled with a rich tapestry of multi-sensory signals. We interact with the world through our inherent multimodal perception, relying not only on language but also on auditory and visual cues for a comprehensive understanding of the world. Enabling machines to process such varied forms of information holds great potential for developing enhanced comprehension of the world. Following this line of thought, recent advancements in Large Language Models (LLMs) (Chiang et al., 2023; Touvron et al., 2023a;b; Zhao et al., 2023a) have motivated researchers to expand these models with both visual and audio encoders, *i.e.*, audio-visual LLMs (Zhang et al., 2023a; Han et al., 2024; Su et al., 2023; Han et al., 2023; Zhao et al., 2023c). This integration offers a comprehensive scene understanding, enabling audio-visual LLMs to interpret videos from both visual and auditory information, similar to humans. Despite promising capabilities in video understanding, these models face significant challenges in jointly handling these signals due to varying amounts of information and inherent heterogeneity of the modalities. Specifically, we observe that audio-visual LLMs are prone to hallucinations, as they may overly rely on one modality while neglecting the other.

Hallucinations, as illustrated in Fig. 1-[Left], denote that audio-visual LLMs hear imaginary sounds from visual cues or perceive fake visual events from audio cues. In other words, multimodal signals may lead to *hallucinations caused by cross-interactions between modalities,* referred to as cross-modal driven hallucination. We hypothesize that this is due to the model's inadequate capacity to handle complex multimodal signals and their relationships. These hallucinations result in outputs

---
*Equal contribution.

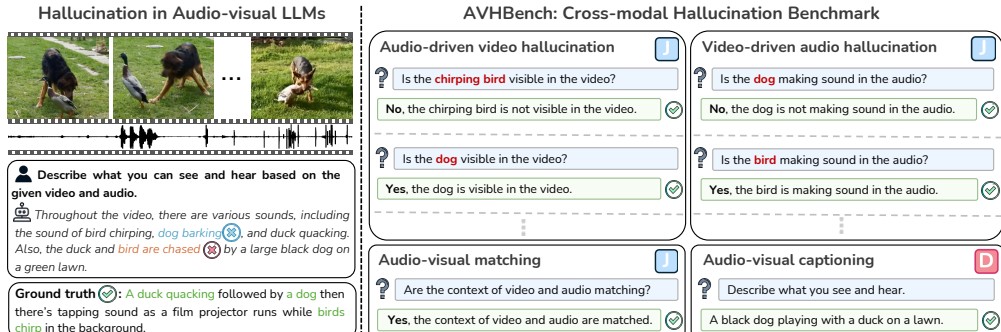

**Figure 1: Cross-modal hallucinations in audio-visual LLMs.** Audio-visual LLMs tend to hallucinate by perceiving non-existent sounds when presented with visual signals (blue event) or imagining non-existent visual events when given an audio signal (orange event), as illustrated on the left. To comprehensively assess these phenomena, we propose AVHBench (depicted on the right), comprising 4 different tasks including the judgment (J) and description (D) tasks. The red objects/event+object in the questions are queried to evaluate the model's perception and robustness against audio-visual hallucinations.

that do not accurately represent the information presented in the input videos and audios. Therefore, assessing these undesirable hallucinations is essential for identifying potential issues, serving as an initial step toward building robust models against hallucinations, and enhancing video understanding.

Despite its importance, there is currently no benchmark to assess audio-visual LLMs' true understanding capabilities with combined audio and visual information. Previous works (Li et al., 2023b; Hu et al., 2023b; Lovenia et al., 2023; Nishimura et al., 2024; Wang et al., 2023a; Liu et al., 2023a) primarily focus on assessing hallucinations within either the audio or visual modality, but not both. For instance, POPE (Li et al., 2023b) and CIEM (Hu et al., 2023b) construct a Question-and-Answer (QnA) format with binary (Yes/No) answers to evaluate models' ability to discern the presence or absence of objects in visual inputs, while Nishimura *et al.* (Nishimura et al., 2024) analyze audio hallucinations in models using pre-trained audio-text models. Although these assessments may systematically measure the level and type of hallucinations in multimodal LLMs, they fall short of evaluating the cross-modal driven hallucinations, as they benchmark within a single modality. However, the true intention of a multimodal hallucination benchmark, specifically audio-visual hallucinations, should involve assessing joint audio-visual inputs and their relationships.

In this work, we introduce AVHBench, an audio-visual hallucination benchmark featuring a dataset of 5,302 QnA pairs and 1,106 audio-visual captions across four distinct tasks: Audio-driven Video Hallucination, Video-driven Audio Hallucination, Audio-visual Matching, and Audio-visual Captioning (See Fig. 1-[Right]). The first three tasks are judgment tasks that identify not only hallucinated objects, as in earlier works, but also the events caused by the objects within the given signal. The fourth task, Audio-visual Captioning, assesses the model's ability to accurately describe the audio and visual signals presented. To compile our hallucination dataset, we establish a semi-automatic annotation pipeline that significantly lowers the cost of human labeling while ensuring high-quality annotations. This pipeline operates in two main stages: (1) disentangling audio-visual objects and events from given videos, and (2) generating QnA pairs tailored to each of the four tasks. The output of this pipeline, our AVHBench, is used to analyze the current status of audio-visual LLMs with respect to the phenomenon of cross-modal driven hallucinations.

Our evaluation results on AVHBench reveal that current audio-visual LLMs are prone to both audio-driven and video-driven hallucinations. We observe that these models generally perform better with unimodal signals or even with text-only inputs. This implies that the models' limited capacity to handle complex multimodal signals may be a potential factor in these hallucinations. Based on insights from our hallucination analysis, we ablate the model with a simple training method using an annotation-enriched training dataset generated by our annotation pipeline. Our findings indicate that Low-Rank Adaptation (LoRA) fine-tuning, coupled with enhanced feature alignment, significantly improves performance. This suggests that refining the attention mechanism of LLMs with audio-visual signals may be one potential approach to enhance the robustness of existing audio-visual LLMs against cross-modal hallucinations. Our key contributions are as follows:

- Proposing `AVHBench`, the first benchmark specifically designed to assess cross-modal hallucinations in audio-visual LLMs, with proposing a semi-automatic annotation pipeline that reduces manual labeling costs while ensuring high-quality annotations.

- Analyzing the presence of cross-modal hallucinations and investigating their potential causes using our proposed `AVHBench` on six recent audio-visual LLMs.

- Demonstrating insights for enhancing the robustness of audio-visual LLMs against cross-modal hallucinations, with improved feature alignment and capacity for handling multimodal signals.

## 2 RELATED WORK

**Multimodal Large Language Models (MLLMs).** Drawing inspiration from the remarkable language abilities of Large Language Models (LLMs) (Achiam et al., 2023; Touvron et al., 2023a;b), researchers have expanded into the field of Multimodal Large Language Models (MLLMs) to enhance performance across diverse multimodal tasks. Primarily, MLLMs have achieved significant progress in integrating vision modalities (Chen et al., 2023a; Dai et al., 2023; Liu et al., 2023b; Peng et al., 2023; Zhang et al., 2023b), *i.e.*, visual LLMs, such as image and video, by attaching visual perception models to LLMs, which possess powerful understanding and reasoning abilities. In a similar way, other types of MLLMs have developed, *e.g.*, audio (Gong et al., 2024; Deshmukh et al., 2023), 3D point cloud (Xu et al., 2023), user interface screens (You et al., 2024), scientific diagrams (Hu et al., 2023a), and fMRI signals (Xu et al., 2023). In this work, we focus on audio-visual LLMs (Su et al., 2023; Han et al., 2023; Zhang et al., 2023a; Zhao et al., 2023c; Panagopoulou et al., 2023; Zhao et al., 2023b), particularly those that incorporate video input, since they are favorable to modeling the multisensory nature of human perception. These models are also typically designed by integrating video and audio encoders with an LLM through the token interface, akin to other MLLMs.

**Hallucinations in MLLMs.** A significant issue with MLLMs is hallucination (Gunjal et al., 2024; Li et al., 2023b; Liu et al., 2024; Wang et al., 2023b; Lovenia et al., 2023; Hu et al., 2023b; Wang et al., 2023a; Hyeon-Woo et al., 2024; Ye-Bin et al., 2024), where the model generates text responses not accurately reflecting the input signal but relying on its internal knowledge (preconception) of the LLM. This tendency to disregard the input signal can lead to adverse outcomes in various real-world applications. To evaluate and find clues of these phenomena, numerous studies have been proposed. One important line of research focuses on hallucination discrimination, aiming to assess MLLMs' ability to identify hallucinations through judgment tasks, such as QnA evaluation. In the context of the visual LLMs, POPE (Li et al., 2023b) and CIEM (Hu et al., 2023b) design binary (Yes/No) questions concerning the presence of objects in images, while NOPE (Lovenia et al., 2023) introduces a Visual Question Answering (VQA) based method to evaluate MLLMs in discerning the absence of objects in visual queries, where correct responses require negative statements. Another branch, such as AMBER (Wang et al., 2023a) and GAVIE (Liu et al., 2023a), focuses on assessing generative hallucinations by prompting models to produce open-ended captions, *i.e.*, description task. However, these works primarily evaluate the hallucination issues related only to the visual modality with the visual LLMs. In this work, our primary objective is to broaden the scope of hallucination evaluation to include simultaneous audio-visual and text inputs. Specifically, we introduce a comprehensive benchmark, `AVHBench`, which includes both judgment and description tasks to assess the perception and comprehension capabilities of audio-visual LLMs against cross-modal driven hallucinations.

## 3 AVHBENCH: CROSS-MODAL HALLUCINATION EVALUATION BENCHMARK

Our goal is to assess the perception and comprehension capabilities of audio-visual LLMs. We particularly aim to analyze the stability of existing models toward cross-modal driven hallucinations and provide insights for developing more robust models with enhanced audio-visual comprehension. As existing benchmarks predominantly focus on evaluating visual hallucinations, we introduce `AVHBench`, designed to evaluate hallucinations in combined audio and visual signals.

**Overview of AVHBench.** We construct 4 different tasks that are fundamental for validating the perception and comprehension of audio-visual LLMs. Inspired by hallucination benchmarks in the vision-language domain (Li et al., 2023b; Lovenia et al., 2023; Hu et al., 2023b), our proposed `AVHBench` includes both *judgment* and *description* hallucination tasks (Liu et al., 2024). Judgment tasks assess the model's ability to identify hallucinated objects and events in the given audio-visual

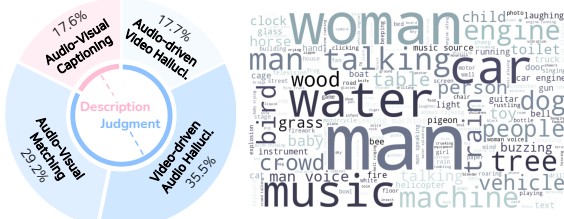

| Task Type | Task | # QnA pairs | # Yes | # No | # captions |
|---|---|---|---|---|---|
| | A→V | 1,136 | 568 | 568 | - |
| Judgment | V→A | 2,290 | 1,145 | 1,145 | - |
| | A-V Mat. | 1,876 | 938 | 938 | - |
| Description | A-V Cap. | - | - | - | 1,106 |
| | Total | 5,302 | 2,651 | 2,651 | 1,106 |

**Figure 2: Dataset statistics.** Our `AVHBench` dataset comprises 2,136 videos featuring 4 different tasks, including 3 judgment tasks and 1 description task. A→V, V→A, A-V Mat., and A-V Cap. denote Audio-driven Video Hallucination, Video-driven Audio Hallucination, Audio-visual Matching, and Audio-visual Captioning respectively. Our dataset contains a total of 5,302 QnA pairs, evenly distributed between yes and no answers, along with 1,106 audio-visual captions.

signals or determine whether those signals correspond to each other. These tasks are structured as binary (Yes/No) questions for intuitive evaluation, suggested in the prior works as a common form. In `AVHBench`, three tasks are categorized into judgment tasks: *Audio-driven Video Hallucination*, *Video-driven Audio Hallucination*, and *Audio-visual Matching*. The description task evaluates whether the model fails to accurately depict the given audio and visual signals. In particular, we focus on *Audio-visual Captioning*, evaluating whether the model outputs hallucinated description when reasoning about the given audio-visual signals. Overall, `AVHBench` comprises 2,136 videos, with 5,302 QnA pairs and 1,106 audio-visual captions. The dataset statistics are summarized in Fig. 2, and more details about the statistics can be found in Appendix D.7.

## 3.1 TASK DEFINITION

We provide detailed descriptions and question formats for 4 different tasks below.

**Audio-driven Video Hallucination.** This task assesses whether the audio signal can cause the model to hallucinate visual objects or events. For example, we evaluate whether the model perceives a certain object as being seen, even though it is not visible but only making sound. The question format is "Is {`object/event+object`} visible in the video?" and the {`object/event+object`} can be either positive, audible in the video, or negative, audible but not visible in the video.

**Video-driven Audio Hallucination.** Conversely, this task evaluates whether the model's auditory perception is influenced by the given visual signal. Although a certain object may exist silently, the model may hallucinate that the object is making sound. The question format is "Is {`object`} making sound in the audio?" and the {`object`} can be either positive, making sound and visible in the video, or negative, existing in the video but not making any sound.

**Audio-visual Matching.** This task evaluates the model's ability to recognize the correspondence between audio and visual signals, determining its accuracy in associating related signals. The question is "Are the context of video and audio matching?". As the paired video and audio naturally match, we randomly swap the original audio with audio from different videos to create a negative pair.

**Audio-visual Captioning (description task).** This task assesses the model's ability to accurately describe combined audio and visual signals. The question is "Describe what you see and hear," where the model is prompted to generate a caption based on the provided audio-visual signals.

Refer to Fig. 1-[Right] for example QnAs related to each task; the summarized template for each task is presented in the Appendix D.2.

## 3.2 DATASET CONSTRUCTION PIPELINE

We design a dataset construction pipeline with automated procedures to minimize human labor. The entire pipeline, depicted in Fig. 3, consists of two main stages: (*Stage 1*) disentangling audio-visual objects and events from given videos, and (*Stage 2*) generating QnA pairs on 4 different tasks.

**Source video collection.** One challenge in constructing the benchmark is collecting the source videos and annotating fine-grained details for both video and audio. To address this, we repurpose existing datasets, namely VALOR (Chen et al., 2023b) and AudioCaps (Kim et al., 2019), leveraging

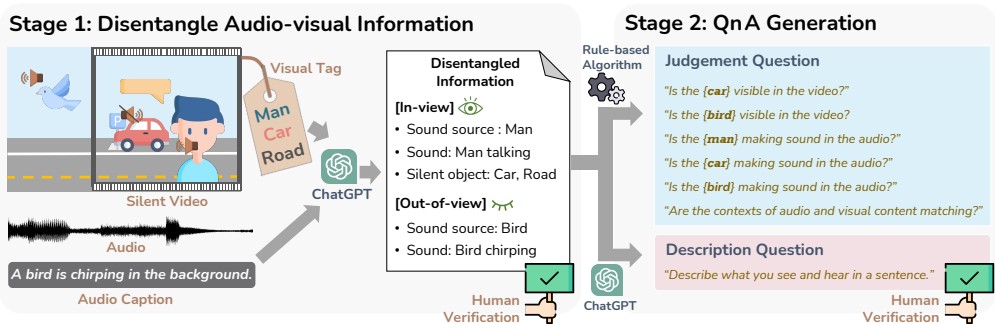

**Figure 3: Dataset construction pipeline.** To design a comprehensive hallucination benchmark, we devise a dataset construction pipeline with automated procedures, consisting of two main stages: Stage 1 involves disentangling audio-visual information, and Stage 2 focuses on Question-and-Answer (QnA) generation for four different tasks. At the end of each stage, we verify and correct the automatically generated outputs by employing a minimal number of human annotators.

their videos and annotations. VALOR contains 32K environmental videos with audio-visual captions. AudioCaps, primarily used for the audio captioning task, features around 50K in-the-wild video. Since both datasets offer either audio-visual or audio captions, we directly utilize their test split for constructing our benchmark.

However, not all source videos consistently contain diverse audio-visual events. In other words, not every video includes out-of-view sounds or multiple events occurring simultaneously. To further facilitate studies on audio-visual comprehension, we have created more challenging synthetic videos by swapping the audio from one video with another. These synthetic videos effectively introduce natural mismatches between audio and visual signals, posing significant challenges for models to accurately discern each signal. In total, our comprehensive test and validation sets comprise 1,106 real and 1,030 synthetic source videos.

**Stage 1: Disentangling audio-visual information.** The goal of this stage is to disentangle audio-visual information from paired video and audio (see Fig. 3-[Stage 1]). Specifically, we separate visual and audio cues into distinct categories: (a) *In-view sound source* indicates visible and audible objects, (b) *In-view silent object* indicates visible but not audible objects, (c) *Out-of-view sound source* indicates audible but not visible objects, (d) *In-view sound* indicates visible and audible sound events, and (e) *Out-of-view sound* indicates audible but not visible sound events.

Disentangling such information directly from video and audio is challenging, thus we rely on the fine-grained annotated captions provided by the VALOR and AudioCaps datasets. However, compared to audio events, visual objects and events appear more frequently and may not be fully captured in existing captions. Therefore, we utilize an off-the-shelf visual tagging model, RAM++ (Zhang et al., 2023c), to extract visual tags from the video. With both visual tagging and text descriptions (either audio or audio-visual captions), we input these metadata into ChatGPT (particularly GPT4 (Achiam et al., 2023)) to disentangle the audio-visual information. Leveraging ChatGPT to assist in constructing the dataset has proven to be effective in previous works (Hyun et al., 2024; Gong et al., 2024; Liu et al., 2023b). We prompt ChatGPT with human-annotated few-shot examples to help it understand the task and perform disentanglement. Example of the Stage 1 outputs, *i.e.*, disentangled audio-visual information, is illustrated in Fig. 3 and Appendix D.6. More details on the prompts for ChatGPT can be found in Appendix D.1.

**Stage 2: Question-and-Answer (QnA) generation for 4 different tasks.** After disentangling the audio-visual information, we use this information to construct QnA pairs for our benchmark, as depicted in Fig. 3-[Stage 2]. For the judgment tasks, we employ a rule-based algorithm to directly utilize the disentangled audio-visual information and form fixed question formats of Yes/No QnA pairs, as describe in Sec. 3.1. For the description task, we once again utilize ChatGPT to take the disentangled audio-visual information and the existing captions (either audio or audio-visual captions) to automatically generate a single sentence of ground truth audio-visual caption. Although the VALOR dataset already provides audio-visual captions, we refine those to be more readable with ChatGPT. Examples of how to use the outputs of Stage 1 to generate questions for each judgment and description task are showcased in Fig. 3. The output samples of Stage 2 are shown in Appendix D.6.

**Human verification.** Although the automated pipeline produces high-quality annotations, we further employ human annotators at the end of each stage to verify the automatically generated outputs. In Stage 1, annotators ensure that audio-visual information is accurately disentangled, correcting any inaccuracies they find. In Stage 2, they verify that the QnA pairs correctly correspond to the given audio and visual signals, editing incorrect answers or discarding ambiguous samples. With minimal human labeling involved, the final dataset becomes sufficiently clean for benchmarking purposes. Additional details about human verification and quality controls are in Appendix D.3 and D.4.

## 4 CROSS-MODAL HALLUCINATION EVALUATION

In this section, we begin by outlining the evaluation setup, detailing the baseline models used, and the evaluation metrics. We then present a thorough analysis of the baseline models' performance on our proposed benchmark, AVHBench, providing insights into potential improvements to enhance their robustness against cross-modal hallucinations.

### 4.1 EVALUATION SETTINGS

**Baseline models.** AVHBench is evaluated on six different recent audio-visual LLMs, mostly referred to in current literature, and capable of processing both audio and visual signals simultaneously. The baseline models are X-InstructBLIP (Panagopoulou et al., 2023), ImageBind-LLM (Han et al., 2023), Video-LLaMA (Zhang et al., 2023a), ChatBridge (Zhao et al., 2023c), PandaGPT (Su et al., 2023), and OneLLM[1] (Han et al., 2024). Each model is configured with a temperature setting of 0 to ensure deterministic output. Given the zero-shot nature of the evaluation, we conduct inference with each model over five trials if a model fails to provide a definitive "Yes" or "No" answer. Additional details and information on prompting can be found in Appendix E.

**Evaluation metrics.** For judgment tasks, we utilize accuracy, precision, recall, and F1 score as the evaluation metrics, by following the conventional metrics used for assessing vision-language hallucinations (Li et al., 2023b; Hu et al., 2023b). Precision and recall indicate the ratios of correctly answered questions with "Yes" or "No" responses, respectively. F1 score is computed by combining the results of precision and recall. We further report the models' "Yes" answer ratio as a reference for analyzing their behavior, with the yes and no answer ratios in our dataset balanced, as shown in Fig. 2. For the description task, we evaluate using widely used captioning metrics, including METEOR (Banerjee & Lavie, 2005) and CIDEr (Vedantam et al., 2015). Moreover, we employ the GAVIE (Liu et al., 2023a) metric, known for its effectiveness in measuring the accuracy of hallucinated captions using GPT4 (Achiam et al., 2023).

### 4.2 EVALUATION RESULTS AND ANALYSIS

Based on our proposed AVHBench, we conduct evaluations to assess the level of cross-modal hallucinations in audio-visual LLMs, analyze the potential cause for such phenomena, and explore the promising solutions. Our extensive analysis aims to address four key questions.

**1) Are audio-visual LLMs robust to cross-modal hallucinations?**

To investigate this, we conduct evaluations with baseline models on four different tasks in our AVHBench. The baseline models are provided with multimodal inputs (audio and video simultaneously, see Fig. 4 (a)) and asked to answer questions. The results are summarized in Table 1. In the

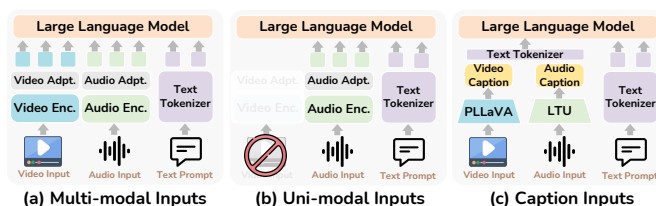

**Figure 4: Input modality types.**

Audio-driven Video Hallucination task, most baseline models perform near or slightly better than random chance, which is 50% accuracy. Furthermore, these models often exhibit overconfidence

---

[1]The OneLLM model that we used for evaluation is not jointly trained as a multimodal model, as reported in https://github.com/csuhan/OneLLM/issues/21, and thus may not perform favorably.

**Table 1: Evaluation results on `AVHBench`.** We evaluate various audio-visual LLMs on our proposed `AVHBench`. Acc. denotes the accuracy and Yes (%) is the proportion of "Yes" answers among total responses. **Bold** numbers stand for the best, and the random choice is provided as a reference.

| Model | Audio-driven Video Hallucination | | | | | Video-driven Audio Hallucination | | | | |
|---|---|---|---|---|---|---|---|---|---|---|
| | Acc. (↑) | Precision (↑) | Recall (↑) | F1 (↑) | Yes (%) | Acc. (↑) | Precision (↑) | Recall (↑) | F1 (↑) | Yes (%) |
| X-InstructBLIP | 18.1 | 16.0 | 15.0 | 15.5 | 46.9 | 16.3 | 14.5 | 38.5 | 21.1 | 77.0 |
| ImageBind-LLM | 50.3 | 50.2 | 87.1 | 63.7 | 86.7 | 50.0 | 50.0 | 99.3 | 66.5 | 99.3 |
| Video-LLaMA | 50.1 | 50.1 | **100** | 66.7 | 99.9 | 50.2 | 50.2 | **100** | 66.9 | 100 |
| ChatBridge | 52.9 | **70.9** | 52.9 | 48.9 | 77.6 | 32.8 | **60.0** | 32.8 | 39.8 | 14.8 |
| PandaGPT | **58.5** | 55.3 | 91.1 | **68.8** | 82.3 | **61.3** | 57.4 | 86.6 | **69.1** | 75.5 |
| OneLLM | 53.7 | 58.6 | 64.8 | 49.8 | 63.1 | 44.3 | 50.2 | 39.4 | 49.8 | 55 |
| Random Choice | 50.0 | 50.0 | 50.0 | 50.0 | 50.0 | 50.0 | 50.0 | 50.0 | 50.0 | 50.0 |

| Model | Audio-visual Matching | | | | | Audio-visual Captioning | | |
|---|---|---|---|---|---|---|---|---|
| | Acc. (↑) | Precision (↑) | Recall (↑) | F1 (↑) | Yes (%) | METEOR (↑) | CIDEr (↑) | GAVIE-A (↑) |
| X-InstructBLIP | 15.1 | 18.6 | 18.9 | 18.8 | 52.6 | 6.10 | 3.40 | 2.83 |
| ImageBind-LLM | 50.0 | 50.0 | **100** | **66.7** | 100 | 11.5 | 16.0 | 3.35 |
| Video-LLaMA | 50.0 | 50.0 | **100** | **66.7** | 100 | **14.0** | 9.5 | 2.29 |
| ChatBridge | 29.9 | 48.3 | 29.9 | 33.9 | 13.0 | 13.7 | **33.1** | **4.69** |
| PandaGPT | 51.2 | 53.6 | 18.1 | 27.0 | 16.8 | 11.7 | 14.1 | 2.70 |
| OneLLM | **60.1** | **67.7** | 61.9 | 64.6 | 53.9 | 5.41 | 28.8 | 1.47 |
| Random Choice | 50.0 | 50.0 | 50.0 | 50.0 | 50.0 | - | - | - |

**Table 2: Evaluation on unimodal hallucinations.** We evaluate baseline models on Audio-driven Video and Video-driven Audio Hallucination tasks by feeding unimodal input to the models, thereby formulating the task into assessing unimodal hallucinations, such as Video Hallucination and Audio Hallucination.

| Model | Video Hallucination | | | | | Audio Hallucination | | | | |
|---|---|---|---|---|---|---|---|---|---|---|
| | Acc. (↑) | Precision (↑) | Recall (↑) | F1 (↑) | Yes (%) | Acc. (↑) | Precision (↑) | Recall (↑) | F1 (↑) | Yes (%) |
| X-InstructBLIP | 46.0 | 47.8 | 90.8 | 62.7 | 94.8 | 34.4 | 37.7 | 46.4 | 41.6 | 61.9 |
| ImageBind-LLM | 52.6 | 51.6 | 84.8 | 64.1 | 82.2 | 50.8 | 50.4 | **98.7** | 66.7 | 97.9 |
| Video-LLaMA | 50.0 | 50.0 | **100** | 66.7 | 100 | 58.4 | 55.7 | 93.3 | **69.8** | 86.2 |
| ChatBridge | 56.5 | 71.2 | 56.5 | 51.9 | 80.1 | 52.6 | 63.9 | 56.7 | 52.6 | 31.0 |
| PandaGPT | 65.0 | 60.1 | 89.2 | 71.8 | 74.2 | **64.1** | 60.4 | 81.6 | 69.4 | 67.5 |
| OneLLM | **76.5** | **75.2** | 79.4 | **77.3** | 53.1 | 59.6 | **73.6** | 55.8 | 63.5 | 47.6 |
| PLLaVA (Video-only LLM) | 81.5 | 77.9 | 87.8 | 82.6 | 56.3 | - | - | - | - | - |
| LTU (Audio-only LLM) | - | - | - | - | - | 67.2 | 63.9 | 84.2 | 72.7 | 68.3 |

(or vice versa), resulting in high recall values despite poor performance. In the Video-driven Audio Hallucination task, the overall performance of the baseline models degrades further. This indicates that audio-visual LLMs are relatively more prone to audio hallucinations than to video hallucinations. Similar trends are observed across the baselines in the Audio-Visual Matching task, where models show high confidence levels (or vice versa), despite performing at random chance levels. Among the baseline models, PandaGPT (Su et al., 2023) performs the best in the Audio-driven Video and Video-driven Audio Hallucination tasks, OneLLM (Han et al., 2024) excels in the Audio-Visual Matching task, while ChatBridge (Zhao et al., 2023c) shows favorable performance in the description task. However, overall, existing baseline models struggle with cross-modal hallucinations, showing undesirable performance. In conclusion, *existing audio-visual LLMs are indeed vulnerable to cross-modal hallucinations*.

**2) Is multimodal input really helpful?** Ideally, as multimodal inputs provide more comprehensive information, the model is expected to perform better compared to using unimodal input. However, we find that *multimodal signals confuse the models' perception*, *i.e.*, *evoking cross-modal driven hallucinations*. To demonstrate this, we conduct the Audio-driven Video and Video-driven Audio Hallucination tasks by feeding unimodal (audio or visual input only, see Fig. 4 (b)) signal to the baseline models, as summarized in Table 2. By comparing the results between Table 1 (multimodal input setting) and Table 2 (unimodal input setting), we observe that the performance of the model improves when using unimodal inputs. This indicates that the existing models suffer from cross-modal driven hallucinations when perceiving and processing multimodal signals simultaneously. Similar phenomena have also been discovered in (Shon et al., 2019; Wang et al., 2020; Senocak et al., 2023), indicating that networks processing multimodal signals struggle more to handle both signals, if they are not properly designed to manage the complex relationships between audio and visual signals.

Based on the observations thus far, one might initially perceive that our proposed `AVHBench` is perhaps overly challenging, unachievable or not properly designed for the assessment. However,

**Table 3: Evaluation on cross-modal hallucinations with caption inputs.** We evaluate baseline models on Audio-driven Video and Video-driven Audio Hallucination tasks with multimodal input converted into text descriptions, thereby formulating the task into text-only input format.

| Model | Audio-driven Video Hallucination | | | | | Video-driven Audio Hallucination | | | | |
|---|---|---|---|---|---|---|---|---|---|---|
| | Acc. (↑) | Precision (↑) | Recall (↑) | F1 (↑) | Yes (%) | Acc. (↑) | Precision (↑) | Recall (↑) | F1 (↑) | Yes (%) |
| X-InstructBLIP | 60.7 | 62.3 | 54.4 | 58.1 | 43.6 | 59.8 | 58.0 | 71.4 | 64.0 | 61.5 |
| ImageBind-LLM | 54.3 | 52.3 | **97.7** | 68.1 | 93.3 | 57.8 | 55.3 | **81.7** | 65.9 | 73.8 |
| Video-LLaMA | 67.2 | 76.2 | 67.2 | 71.4 | 43.3 | 62.3 | **69.2** | 62.3 | 65.4 | 40.6 |
| ChatBridge | 67.1 | 65.0 | 67.1 | 65.6 | 22.2 | 61.1 | 66.4 | 61.1 | 62.9 | 40.7 |
| PandaGPT | 66.7 | **84.6** | 40.8 | 55.1 | 24.1 | **67.2** | 66.7 | 68.9 | 67.8 | 51.6 |
| OneLLM | **76.6** | 75.4 | 79.6 | **77.4** | 53.2 | 62.6 | 68.9 | 73.9 | **71.3** | 67.5 |

we find that AVHBench is indeed achievable to some extent. To demonstrate this, we utilize existing modality-specific video LLM and audio LLM to conduct experiments on Audio-driven Video Hallucination and Video-driven Audio Hallucination tasks, respectively. More specifically, we use PLLaVA (Xu et al., 2024), a video-only LLM, and LTU (Gong et al., 2024), an audio-only LLM. As shown in Table 2, these models significantly outperform the existing audio-visual LLMs. These results verify that our benchmark is sufficiently achievable and suggest that audio-visual LLMs have considerable room for improvement.

**3) What may be the potential problem for the cross-modal hallucinations?** We suspect that the *LLMs' limited capacity to handle complex multimodal signals is one reason for hallucinations*. To test this hypothesis, we reformulate the tasks into a text-only problem, providing more familiar inputs for the LLMs to process (see Fig. 4 (c)). Specifically, the input signals are converted into text descriptions (captions) using LTU (Gong et al., 2024) for audio input and PLLaVA (Xu et al., 2024) for video input, respectively[2]. These captions are then concatenated and fed directly to the LLM component of audio-visual LLMs to answer to the questions without any further training.

We evaluate the performance of this text-only input approach on the Audio-driven Video and Video-driven Audio Hallucination tasks, as summarized in Table 3. By comparing the results from Table 1 (multimodal input) and Table 3 (text-only input), using multimodal captions outperforms direct multimodal inputs. Specifically, PandaGPT (Su et al., 2023), Video-LLaMA (Zhang et al., 2023a), and OneLLM (Han et al., 2024) show significant improvement in the Audio-driven Video Hallucination task and Video-driven Audio Hallucination task, respectively. Through these results, we hypothesize that the instability of the models against cross-modal hallucinations may be due to the LLMs' inability to perceive the multimodal signals effectively and learn the relationships between these signals, such as alignment, binding, or distinction.

**4) Can audio-visual LLMs be improved against cross-modal hallucinations?** Yes, audio-visual LLMs can be enhanced by applying simple training methods if carefully designed. We observe that several existing audio-visual LLMs, *e.g.*, PandaGPT (Su et al., 2023) and Video-LLaMA (Zhang et al., 2023a), are trained without incorporating any audio data. In other words, despite being audio-visual models, they lack joint training using both audio and visual inputs, which may cause a weak alignment between audio representations and the LLMs. Based on these observations, we attempt to improve existing models from two perspectives: (1) enhancing the alignment of audio features, and (2) Low-Rank Adaptation (LoRA) (Hu et al., 2022) fine-tuning. In these experiments, we select Video-LLaMA as the baseline model[3].

**Audio feature alignment.** To enhance the alignment between audio features and LLM, we pre-train Q-former (Li et al., 2023a) and a linear layer that links the frozen audio encoder and LLM, while keeping the video part fixed. We utilize the audio-text paired datasets, such as AudioCaps (Kim et al., 2019), Clotho (Drossos et al., 2020), and WavCaps (Mei et al., 2023), for training.

**LoRA FT.** For this, we collect a large, annotation-enriched audio-visual dataset using the semi-automatic pipeline described in Sec. 3.2, without involving any human verification. This dataset contains 10,327 videos with 87,624 QnA pairs, collected from the training split of the VALOR (Chen et al., 2023b) and Audiocaps (Kim et al., 2019) datasets. We apply LoRA fine-tuning to Video-LLaMA using this acquired dataset, while the audio and video encoders are kept frozen.

---

[2]Note that these captions are generated, not ground truth, and therefore noisy.
[3]More details about training can be found in Appendix F.

**Table 4: Enhancing robustness against cross-modal hallucinations.** Align. represents enhancing audio feature alignment with LLM, and FT denotes LoRA fine-tuning on the training split of the `AVHBench` dataset. multi. and uni. stands for multimodal inputs and unimodal input, respectively.

| Align. | FT | A→V (multi.) | A→V (uni.) | V→A (multi.) | V→A (uni.) | A-V Mat. | Audio-visual Captioning | | |
|--------|-----|------|------|------|------|------|------|------|------|
| | | | | | | | METEOR (↑) | CIDEr (↑) | GAVIE-A (↑) |
| - | - | 50.1 | 50.0 | 50.2 | 55.7 | 50.0 | **14.0** | 9.5 | 2.29 |
| ✓ | - | 52.8 | 50.4 | 58.1 | 63.5 | 51.3 | 9.5 | 18.9 | 3.49 |
| - | ✓ | 79.1 | 84.0 | 76.6 | 80.6 | 50.8 | 11.9 | 33.1 | 3.54 |
| ✓ | ✓ | **83.9** | **85.2** | **77.3** | **81.1** | **55.6** | 12.2 | **35.6** | **3.82** |

**Table 5: Generalized improvements on other benchmarks.** We test the generalization performance of our ablated variants on VAST captioning dataset (Chen et al., 2023c) and an audio-visual joint instruction dataset, AVinstruct (Ye et al., 2024).

| Align. | FT | VAST | | | AVinstruct | | | | |
|--------|-----|------|------|------|------|------|------|------|------|
| | | METEOR (↑) | CIDEr (↑) | GAVIE-A (↑) | METEOR (↑) | CIDEr (↑) | ROUGE-L (↑) | BLEU-4 (↑) | Acc. (%) |
| - | - | 18.2 | 0.2 | 4.04 | 45.9 | 14.5 | 35.3 | 12.8 | 43.6 |
| ✓ | - | 19.2 | 20.7 | 3.68 | 42.2 | 27.1 | 41.5 | 14.9 | 52.6 |
| - | ✓ | 18.7 | 13.4 | 2.58 | 53.5 | 76.4 | 52.3 | 25.1 | 44.2 |
| ✓ | ✓ | **22.1** | **47.6** | **5.09** | **58.1** | **102.0** | **55.8** | **28.5** | **57.8** |

Table 4 presents the results of the ablation study on these training schemes. Improving audio alignment, in comparison to the original Video-LLaMA, results in enhanced performance on the Video-driven Audio Hallucination task, as the model's audio perception becomes stronger and more reliable. Moreover, LoRA fine-tuning with our dataset shows significant improvement, indicating that exposing the model to learn appropriate attention from both audio and visual cues is a crucial training step for enhancing robustness against hallucinations. Combining enhanced audio alignment with LoRA fine-tuning achieves further performance gains across all tasks. We also evaluate the ablated variants on the VAST captioning dataset (Chen et al., 2023c) and AVinstruct (Ye et al., 2024), an audio-visual joint instruction dataset. For AVinstruct, we provide results for both captioning metrics (Banerjee & Lavie, 2005; Vedantam et al., 2015; Lin, 2004; Papineni et al., 2002) and the accuracy of multiple-choice (*i.e.*, closed-ended) questions. Table 5 demonstrates the generalized improvements in zero-shot scenarios by enhancing robustness against cross-modal hallucinations.

Finally, Fig. 5 shows the qualitative results of the existing audio-visual LLMs (Su et al., 2023; Zhang et al., 2023a; Zhao et al., 2023c) and our final model on `AVHBench`. Several existing audio-visual LLMs perceive imaginary "bell" from audio cues or fake sound of "screen" from visual cues, while our final model does not suffer from these cross-modal hallucinations. Moreover, the existing audio-visual LLMs struggle to faithfully represent the provided audio and visual signals, often generating objects or events that do not actually exist. Refer to Appendix B for more qualitative results.

## 5 LIMITATIONS AND FUTURE WORK

As a preliminary study on revealing audio-visual hallucinations, it is important to acknowledge the limitations of this work. Given the complexity of multimodal signals and their relationships, including the dynamics of sound emission and localization within visual events or objects, our evaluation may not encompass all possible scenarios. Therefore, there is a need for a more diverse, complex, and larger dataset with more comprehensive tasks. This expansion could be achieved by acquiring additional data from a range of in-the-wild sources, such as egocentric videos, or by synthesizing diverse scenes using simulation platforms.

Additionally, our primary focus is on assessing how one modality might influence the perception of another; consequently, our benchmark may not include more fine-grained evaluations, such as hallucinations caused by multiple objects in the same scene or temporal reasoning within audio-visual captions. We believe that incorporating such complex scenarios and providing appropriate annotations would enhance the dataset's comprehensiveness, enabling finer-grained analyses. For instance, extending the dataset with source video datasets that include temporal QnAs (Li et al., 2022; Yang et al., 2022) could facilitate constructing audio-visual captions with temporal reasoning. Moreover, while our current audio-visual captions effectively captures objects, actions, and sound events within 10-second clips, longer and more complex videos may require extended audio-visual captions to comprehensively capture all presented information.

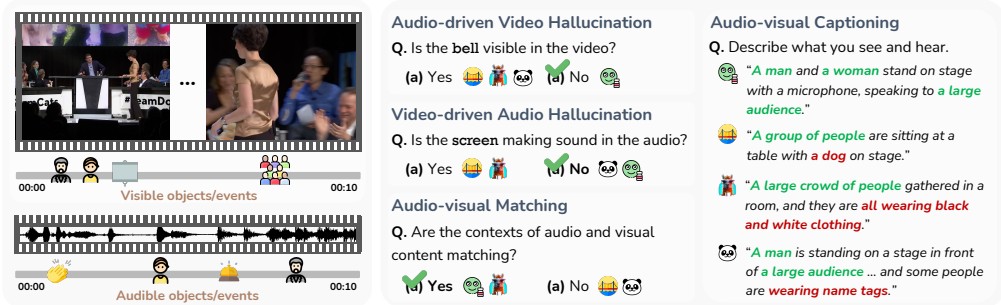

**Figure 5: Qualitative results.** We illustrate visible and audible objects/events in the video (on the left). Green denotes the correct answers and Red denotes the incorrect answers produced by audio-visual LLMs. 🧑‍🔬, 🐼, 🤖, and 😀 stand for our final model, PandaGPT (Su et al., 2023), Video-LLaMA (Zhang et al., 2023a), and ChatBridge (Zhao et al., 2023c), respectively.

Finally, we focus on "environmental sound" rather than speech, which might neglect how the content in speech influences hallucinations in video interpretation. A promising future direction could involve using Automatic Speech Recognition (ASR) to integrate speech content into our dataset construction, potentially exploring how speech influences visual perception hallucinations.

## 6 CONCLUSION

In this paper, we introduce AVHBench, a cross-modal hallucination evaluation benchmark designed to assess the recent audio-visual LLMs. AVHBench contains four fundamental tasks for validating the perception and comprehension capabilities of these models. We have developed a semi-automatic annotation pipeline that significantly reduces human labor while maintaining high-quality annotations of this benchmark. Using this benchmark, we observe that audio-visual LLMs are prone to cross-modal hallucinations and tend to perform better with unimodal or text-only inputs, rather than using multimodal inputs. Based on these analysis, we suggest that enhancing both feature alignment and the capacity to handle multimodal signals could improve the models' robustness against hallucinations.

### ETHICS STATEMENT

The proposed cross-modal hallucination benchmark in this work has undergone a human annotation process, effectively mitigating negative biases through careful human review. Therefore, we do not expect any potential ethical issues related to sensitive information in the dataset. However, since the training set used to improve the stability of cross-modal hallucination is automatically annotated without manual intervention, it may have potential negative biases from the data distribution of the original AudioCaps (Kim et al., 2019) and VALOR (Chen et al., 2023b) datasets. Such biases, including the over-represented urban environments and the under-represented rural settings, may affect the model's reliability and fairness when deployed in various real-world settings. Therefore, users should be aware of the potential for biased outputs and consider these factors when integrating the models into larger systems.

### REPRODUCIBILITY STATEMENT

We introduce the high-level semi-automatic dataset generation pipeline used for AVHBench in Sec. 3.2, which includes details of each annotation stage, the source video datasets utilized (Chen et al., 2023b; Kim et al., 2019), and the off-the-shelf models employed (Achiam et al., 2023; Zhang et al., 2023c). A more detailed description of the pipeline, such as the ChatGPT prompts for audio-visual information disentanglement in Stage 1 and audio-visual caption generation in Stage 2, is provided in Appendix D. Since our dataset requires minimal human annotation, the annotation interface and our approach for annotation quality control are described in Appendix D.3 and D.4. For evaluation, the baseline models and evaluation setups are detailed in Sec. 4.1 and Appendix E, while the training recipe and the dataset used for enhancing the model's stability against cross-modal hallucination are outlined in Sec. 4.2 and Appendix F.

ACKNOWLEDGMENTS

This work was partially supported by IITP grants (No.RS-2023-00225630, Development of Artificial Intelligence for Text-based 3D Movie Generation (15%); No.RS-2021-II212068, Artificial Intelligence Innovation Hub (15%); No. 2022-0-00124, No.RS-2022-II220124, Development of Artificial Intelligence Technology for Self-Improving Competency-Aware Learning Capabilities(15%)), and the National Research Foundation of Korea (NRF) grant (No. RS-2024-00358135, Corner Vision: Learning to Look Around the Corner through Multi-modal Signals (15%)) funded by the Korea government (MSIT). A. Senocak and J. S. Chung were partially supported by the National Research Foundation of Korea (NRF) grant funded by the Korea government (MSIT) (No. RS2023-00212845 (40%)).

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

APPENDIX

In this supplementary material, we provide comprehensive details about the data annotation procedure of our proposed `AVHBench` dataset, the experimental setups, and additional experimental results that were not included in the main paper.

CONTENTS

A    DATA SAMPLES AND SOURCE CODE

We provide the data processing, annotation code, and a subset of our `AVHBench` dataset through the .zip file. The complete set of the `AVHBench` dataset will be released upon acceptance. Additionally, please refer to the datasheet for the `AVHBench` dataset attached as an explicit PDF.

B    ADDITIONAL QUALITATIVE RESULTS

In this section, we include additional qualitative results of the existing audio-visual LLMs and our final model on `AVHBench`. Moreover, we visualize the attention maps of the transformer layers in the LLM from Video-LLaMA (Zhang et al., 2023a) and our final model, to investigate which parts the model focuses on to answer the cross-modal hallucination questions.

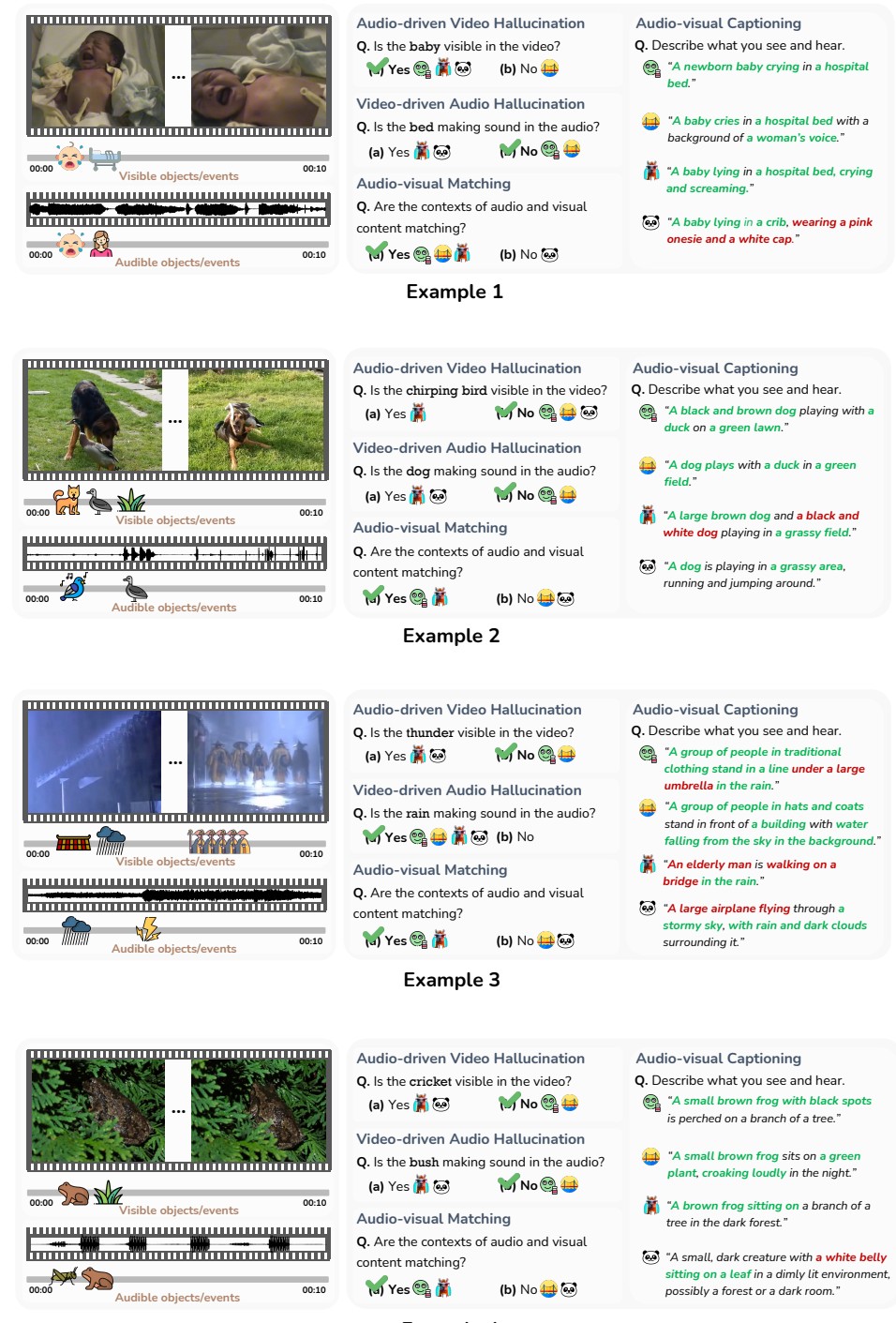

**Figure S1: Additional qualitative examples on AVHBench test dataset.** We illustrate visible and audible objects/events in the given video, with **green** denoting the correct answers and **red** denoting the incorrect ones. The icons in the figure, 🐼, 🧙, 🌅 and 🐸 (Ours) stand for PandaGPT (Su et al., 2023), Video-LLaMA (Zhang et al., 2023a), ChatBridge (Zhao et al., 2023c), and our final model, respectively.

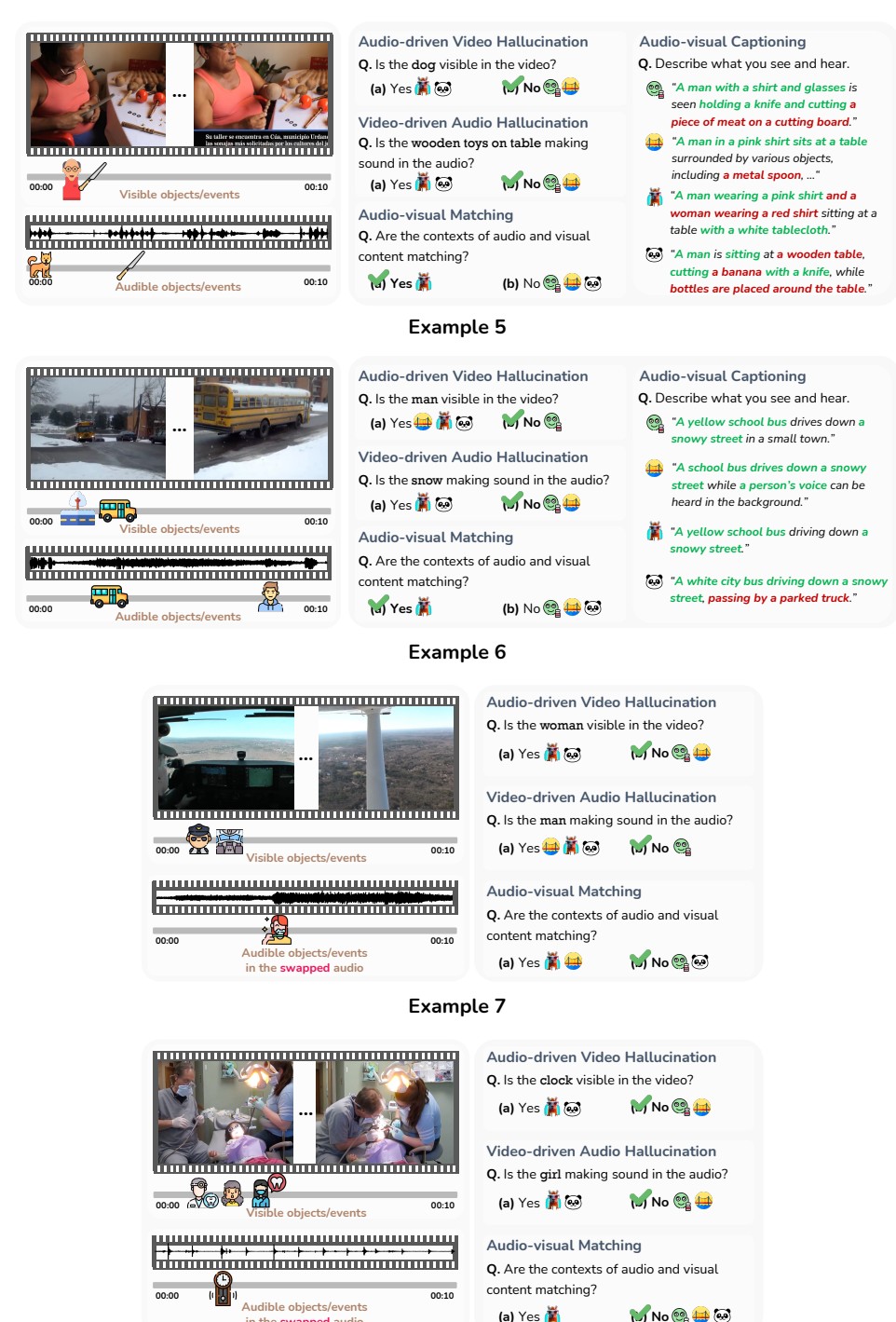

**Figure S2: Additional qualitative examples on AVHBench test dataset.** We illustrate visible and audible objects/events in the given video, with **green** denoting the correct answers and **red** denoting the incorrect ones. The icons in the figure, 🐼, 🦙, 🌉 and 🟢 (Ours) stand for PandaGPT (Su et al., 2023), Video-LLaMA (Zhang et al., 2023a), ChatBridge (Zhao et al., 2023c), and our final model, respectively. The two examples above are for real video samples, while the other ones below are for synthetic (swapped) video samples.

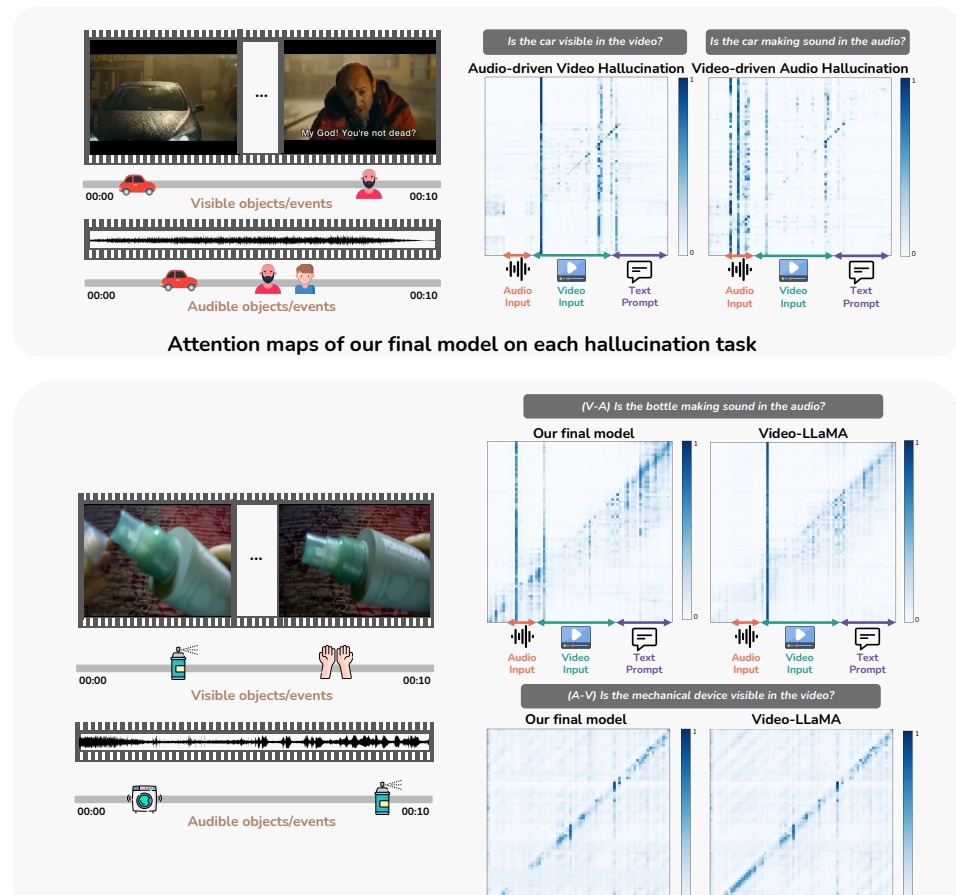

**Figure S3: Attention visualization of the LLM layers.** We visualize the attention maps of the LLM layers to identify which modality the model prioritizes to answer the cross-modal hallucination questions. In the first row, we show the attention maps of our final model on each hallucination task. Additionally, we compare the attention maps of our final model and Video-LLaMA (Zhang et al., 2023a) on each hallucination task. We illustrate visible and audible objects/events in the given video. The darker blue color in the attention maps denotes higher attention scores. We represent the token indices for the audio, video, and text prompt inputs as orange, green, and purple arrows, respectively.

**Qualitative results.** In addition to Fig. 5 in the main paper, we provide additional examples of the predicted results in the hallucination tasks for our final model and the baseline models (Zhang et al., 2023a; Su et al., 2023; Zhao et al., 2023c) (See Example 1-6 for the real video samples and Example 7-8 for the synthetic video samples in Figs. S1 and S2). Note that all real videos inherently have "Yes" as correct answers for Audio-visual Matching task, whereas synthetic videos generated by swapping the audio only have "No" as correct answers. We depict visible and audible objects/events in the video, with **green** representing the correct answers and **red** representing the incorrect ones. The illustrations in the figure, 🐼, 🦙, 🌉 and 🤖 (Ours) denote PandaGPT (Su et al., 2023), Video-LLaMA (Zhang et al., 2023a), ChatBridge (Zhao et al., 2023c), and our final model, respectively. The results indicate that Video-LLaMA tends to consistently answer all questions in the judgment task as "Yes", leading to nearly random performance, as demonstrated in Table. 2 in the main paper. Our final model shows favorable accuracy in three judgment tasks, compared to Video-LLaMA, PandaGPT (Su et al., 2023), and ChatBridge (Zhao et al., 2023c). PandaGPT and Video-LLaMA often represent the objects and events that are not grounded on the given video, while our final model and ChatBridge are less prone to these cross-modal hallucinations when generating the audio-visual captions.

**Table S1: Additional evaluation results on `AVHBench`.** We evaluate various more recent audio-visual LLMs and uinmodal LLMs on our proposed `AVHBench`. In the Audio-driven Video and Video-driven Audio Hallucination, both audio-visual signals are used as inputs for the model. For Video and Audio Hallucinations, only video or audio inputs are used, respectively. Acc. denotes the accuracy, Yes (%) is the proportion of "Yes" answers among total responses, and **bold** numbers stand for the best.

| Model | Audio-driven Video Hallucination | | | | | Video-driven Audio Hallucination | | | | |
|---|---|---|---|---|---|---|---|---|---|---|
| | Acc. (↑) | Precision (↑) | Recall (↑) | F1 (↑) | Yes (%) | Acc. (↑) | Precision (↑) | Recall (↑) | F1 (↑) | Yes (%) |
| Video-LLaMA | 50.1 | 50.1 | **100** | 66.7 | 99.9 | 50.2 | 50.2 | **100** | 66.9 | 100 |
| Video-SALMONN | 78.1 | 74.9 | 84.5 | 79.4 | 56.4 | 65.2 | 62.3 | 76.9 | 68.8 | 61.7 |
| Video-LLaMA2 | 75.2 | 73.6 | 78.7 | 76.1 | 53.6 | **74.2** | **69.4** | 86.6 | **77.0** | 62.4 |
| Gemini-Flash | **83.3** | **85.7** | 81.0 | **83.7** | 47.3 | 63.0 | 57.9 | 94.7 | 71.9 | 81.7 |

| Model | Video Hallucination | | | | | Audio Hallucination | | | | |
|---|---|---|---|---|---|---|---|---|---|---|
| | Acc. (↑) | Precision (↑) | Recall (↑) | F1 (↑) | Yes (%) | Acc. (↑) | Precision (↑) | Recall (↑) | F1 (↑) | Yes (%) |
| Video-LLaMA | 50.0 | 50.0 | **100** | 66.7 | 100 | 58.4 | 55.7 | 93.3 | 69.8 | 86.2 |
| Video-SALMONN | 82.3 | 83.9 | 79.9 | 81.9 | 47.6 | 66.5 | 70.5 | 56.8 | 62.9 | 40.3 |
| Video-LLaMA2 | 80.7 | 87.0 | 72.1 | 78.8 | 41.4 | **79.8** | **81.9** | 76.4 | **79.1** | 46.7 |
| Gemini-Flash | **84.3** | 81.6 | 72.2 | 76.6 | 31.4 | 65.2 | 59.6 | **94.3** | 73.0 | 79.1 |
| PLLaVA | 81.5 | 77.9 | 87.8 | **82.6** | 56.3 | - | - | - | - | - |
| LLaVA-OneVision | 84.0 | **90.9** | 75.5 | 82.5 | 41.5 | - | - | - | - | - |
| LTU | - | - | - | - | - | 67.2 | 63.9 | 84.2 | 72.7 | 68.3 |
| Qwen2-Audio | - | - | - | - | - | 74.4 | 81.3 | 68.8 | 74.5 | 13.6 |

**Attention map visualization.** We visualize the attention maps of the LLM layers to identify which modality the model prioritizes to answer the cross-modal hallucination questions. As illustrated in the left attention map of Fig. S3-[First Row], our final model shows higher attention scores on the video token indices in the Audio-driven Video Hallucination task, where the model should prioritize the visual modality. In contrast, in the Video-driven Audio Hallucination task, our final model exhibits higher attention scores on the audio token indices, indicating that the model appropriately utilizes the audio information (See the right attention map of Fig. S3-[First Row]). Additionally, we compare the attention maps between our final model and Video-LLaMA (Zhang et al., 2023a). As depicted in Fig. S3-[Second Row], our model shows higher attention scores on the audio token indices (Left), whereas Video-LLaMA shows lower scores (Right) in the Video-driven Audio Hallucination task. This suggests that our model has a favorable ability in leveraging audio information for audio-related questions, possibly driven by audio alignment. In Fig. S3-[Third Row], for the Audio-driven Video Hallucination task where the model should not rely on audio information, our model effectively reduces the attention to audio modality. Conversely, Video-LLaMA maintains focus on audio information, with higher attention scores on the audio token indices. This implies that Video-LLaMA may be susceptible to imaginary objects or events in the video, which are driven from the audio modality.

## C  ADDITIONAL QUANTITATIVE RESULTS

We conduct further evaluations on `AVHBench` using more recent audio-visual LLMs, such as Video-SALMONN (Sun et al., 2024), Video-LLaMA2 (Cheng et al., 2024), and GEMINI-Flash (Anil et al., 2023), and unimodal LLMs, such as LLaVA-OneVision (Li et al., 2024) and Qwen2-Audio (Chu et al., 2024). The results are summarized in Table S1, and we also include reference results from Video-LLaMA (Zhang et al., 2023a), PLLaVA (Xu et al., 2024), and LTU (Gong et al., 2024). We generally observe that recent models are improving on our benchmark, for both audio-visual LLMs and unimodal LLMs. GEMINI significantly outperforms other audio-visual LLMs in Audio-driven Video Hallucination. In Video-driven Audio Hallucination, Video-LLaMA2 exhibits strong performance compared to other models. Interestingly, we find that overall model performance improves when using unimodal signals for evaluation as summarized in Video and Audio Hallucination results. Specifically, Video-LLaMA2 demonstrates notable improvements with unimodal signals compared to using both audio-visual signals, supporting discussions in Sec. 4.2 on how cross-modal signals can confuse models and evoke hallucinations. These findings highlight recent enhancements in models to address cross-modal hallucination and also describe the effectiveness of our benchmark in evaluating these advancements.

**Table S2: Prompt for audio-visual information disentanglement.** The provided prompt is fed into GPT4 to disentangle the given content into five different audio-visual information. We manually select a sample video and fill in the corresponding caption and visual tagging information for each video, completing the fields within the prompt template. Note that the caption is provided by the original dataset, AudioCaps (Kim et al., 2019) or VALOR (Chen et al., 2023b), and the visual taggings are extracted from RAM++ (Zhang et al., 2023c) from the video.

---

The "caption" provides the explanation about audio event. The "visual tagging" identifies the observable visual objects or actions. If similar objects from the "caption" are found in the "visual tagging", it is assumed to exist in-view. If not, it is assumed to exist out-of-view. The "information" aims to summarize the inferable audio and visual information: In-view sound source is the object that makes sound and is visible in the scene. In-view sound is the type of sound that the object in the scene makes, In-view silent object is the object that does not make sound but is visible in the scene, Out-of-view sound source is the expected object that makes sound and is not visible in the scene, and Out-of-view sound is the type of sound that the object out of the scene makes.

caption: {caption corresponding to the given video}
visual tagging: {visual taggings corresponding to the given video}
information:
- In-view sound source:
- In-view sound:
- In-view silent object:
- Out-of-view sound source:
- Out-of-view sound:

---

## D    DETAILS ON THE DATA CONSTRUCTION PIPELINE AND STATISTICS

The dataset construction pipeline outlined in the main paper comprises two main stages: (*Stage 1*) Disentangling audio-visual information, and (*Stage 2*) Question-and-Answer (QnA) generation for four different cross-modal hallucination tasks. Stage 1 incorporates the use of GPT4 (Achiam et al., 2023) for generating pseudo annotations, which are then verified by human annotators to ensure the accuracy of the automatically generated outputs. In Stage 2, using the disentangled audio-visual information from Stage 1, QnAs are automatically generated through a rule-based algorithm, and audio-visual captions are produced using GPT4. This stage also involves human verification to ensure the accuracy and quality of the QnAs and captions. The source videos of our proposed AVHBench dataset is based on two widely used audio and video datasets, AudioCaps (Kim et al., 2019) and VALOR (Chen et al., 2023b).

### D.1    GPT4 PROMPTS

Table. S2 illustrates an example GPT4 prompt used for generating pseudo annotations in Stage 1. In Stage 1, visual tags and audio (or audio-visual) captions are fed into GPT4 to disentangle the audio-visual information, including in-view sound sources, in-view sounds, in-view silent objects, out-of-view sound sources, and out-of-view sounds. Given the complexity of this task, we find that providing GPT4 with several human-annotated examples is crucial for enhancing its understanding and performance. To this end, we utilize a 3-shot in-context learning approach by supplying three human-annotated examples to GPT4.

Table. S3 presents an example prompt for GPT4 to generate an audio-visual caption for the description task. We provide both the audio caption and video caption to GPT4, which then merges these captions to produce a comprehensive audio-visual caption. The provided video caption is generated by RAM++ (Zhang et al., 2023c), using the visual tags. Similar to the previous stage, human-annotated examples are provided to improve GPT4's ability to generate accurate and detailed audio-visual captions.

**Table S3: Prompt for the audio-visual caption generation.** The provided prompt is fed into GPT4 to generate audio-visual captions. We manually select a sample video and fill in the corresponding audio and video captions for each video, completing the fields within the prompt template.

| |
|---|
| Based on the given audio and video captions, generate an audio-visual caption in a sentence to reflect both information. Audio caption: {audio caption corresponding to the given audio} Video caption: {visual caption corresponding to the given video} |
| [Example] |
| Based on the given audio and video captions, generate an audio-visual caption in a sentence to reflect both information. **Audio caption**: A man talking as ocean waves trickle and splash while wind blows into a microphone **Video caption**: the bow of a boat navigating in the ocean with large waves, a view from the bow of a boat of a large body of water with waves and clouds, the bow of a sailboat navigating in the ocean with the wind in the water, the bow of a sailboat navigating in the open sea with the wind and waves, the bow of a boat with a rope and large sail navigating in the ocean, sunlight shines down on a body of water as the sun reflects off the ocean, sunlight shines through the clouds in the sky over a body of water with mountains in the distance, the bow of a boat with a rope and sail navigating in the ocean |
| **Generated audio-visual caption**: As the man's voice blends with the sound of ocean waves trickling and splashing, the bow of a sailboat navigates through the open sea with the wind and waves. |

**Table S4: Question templates for each task.**

| Task Type | Task | Question Template | Answer |
|---|---|---|---|
| Judgment | Audio-driven Video Hallucination | Is the {object/event} visible in the video? | Yes/No |
| | Video-driven Audio Hallucination | Is the {object/event} making sound in the audio? | Yes/No |
| | Audio-visual Matching | Are the context of audio and visual content matching? | Yes/No |
| Description | Audio-visual Captioning | Describe what you see and hear. | Free-form |

## D.2  POSITIVE AND NEGATIVE SAMPLES IN QNAS

As summarized in Table. S4, three judgment tasks require both positive and negative samples to construct QnAs. For the Audio-driven Video Hallucination task, we use in-view sound sources as positive objects or events, while out-of-view sound sources serve as negative examples. Moreover, for the Video-driven Audio Hallucination task, in-view sounds and sound sources are considered positive, whereas in-view silent objects are used as negative examples. For the Audio-visual Matching task, as the paired video and audio naturally match, we randomly swap the audio with audio from different videos to create a negative pair.

## D.3  DETAILS ON HUMAN ANNOTATION

Human annotators play a crucial role in validating the outputs of GPT4 in each stage. Figure. S4 displays the interface and instructions for annotators during the verification process for Stage 1. Here, audio-visual information categorized by GPT4 is reviewed; annotators are tasked with checking whether these information are correctly categorized and correcting any errors. Figure. S5 shows the interface and instructions for Stage 2 verification, where the QnA pairs and audio-visual captions are initially generated by a rule-based algorithm and GPT-4, respectively. Similar to Stage 1, annotators verify each QnA pairs, correct any inaccuracies, or discard ambiguous samples to ensure data quality.

## D.4  ANNOTATION QUALITY CONTROL

To ensure high-quality annotations, we establish the qualification criteria for screening annotators as individuals with experience in annotating both video and audio data. In Stage 1 human verification, annotators are tasked with verifying and editing initial pseudo annotations generated by GPT-4. They then manually review these annotations on samples they have not previously worked on, enabling cross-checking to enhance the output quality. In Stage 2 human verification, three annotators are employed to verify the QnA pairs and audio-visual captions. One annotator is asked to check QnAs

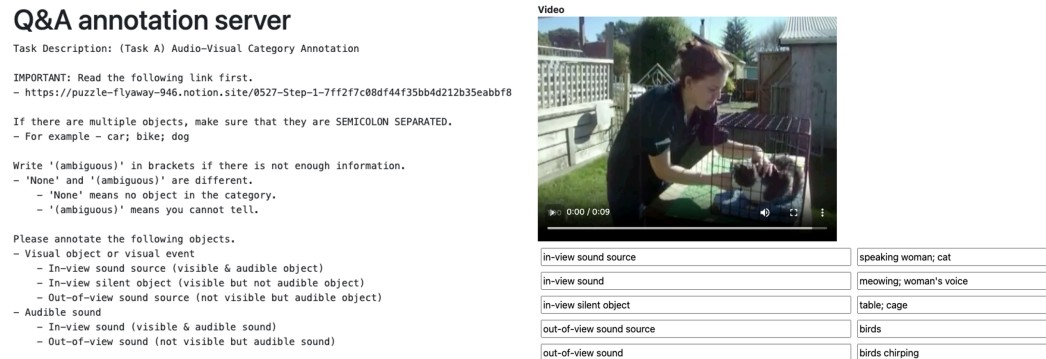

**Figure S4: Stage 1 human annotation interface.** The annotators are asked to read the instruction and watch the video clip. They check whether the information is disentangled correctly and make any necessary corrections.

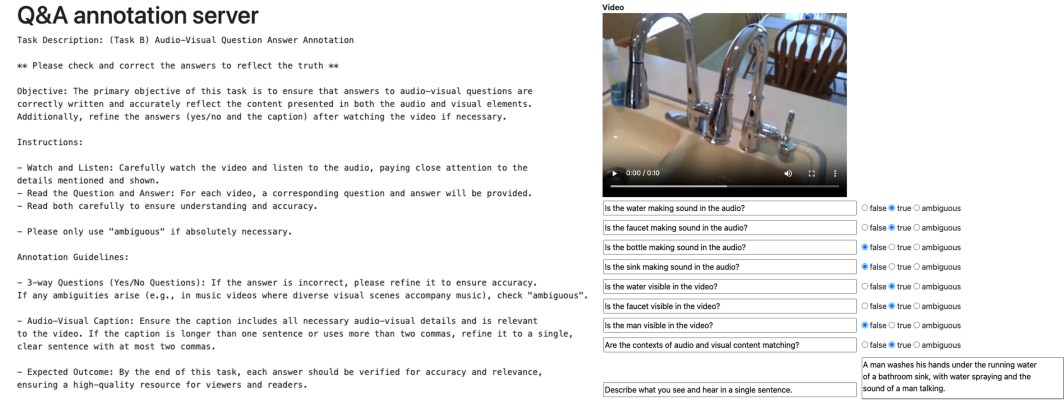

**Figure S5: Stage 2 human annotation interface.** The annotators are asked to read the instruction and watch the video clip. They assess whether the answers correspond to the questions and the audio-visual caption accurately reflects the given video content. If they find any incorrectness, they are asked to correct or discard them.

and audio-visual captions that are entirely initialized with pseudo-annotations. Another annotator performs a similar task, but with approximately 10% of the QnA pairs intentionally made inaccurate, given that the pseudo annotations are fairly accurate. This annotator revises approximately 9.8% of the answers among 10% inaccurate ones. The final annotator annotates the QnAs and audio-visual captions from scratch. Since QnA pairs are categorized into binary, we finalize the response based on the majority vote from the annotators. If a caption is revised by the annotators, the authors manually review the edited versions and select the one that best explains the video. During the human annotations, 94.4% of the QnA pairs in the stage 2 annotations agreed with all three annotators, while 5.6% agreed with two. After these verifications, approximately 20% of the QnA responses are corrected.

### D.5 MODEL-BASED ANNOTATION ERROR STUDY

As automatic annotations rely on off-the-shelf models, initial errors are unavoidable. To address this, human verification plays a crucial role in our semi-automatic dataset construction pipeline for correcting errors that sometimes occur from noisy model outputs. We provide examples of how this process enhances annotation accuracy. For example, in Stage 1, RAM++ (Zhang et al., 2023c), which extracts visual tags from videos, might include noisy tags that misrepresent the audio-visual information. As shown in Fig. S6-(a), despite the absence of a "plane" in the video, it is listed as an in-view silent object because it was mistakenly tagged by RAM++. However, we observe that Stage 1 annotators successfully identify and remove these inaccuracies during the annotation process.

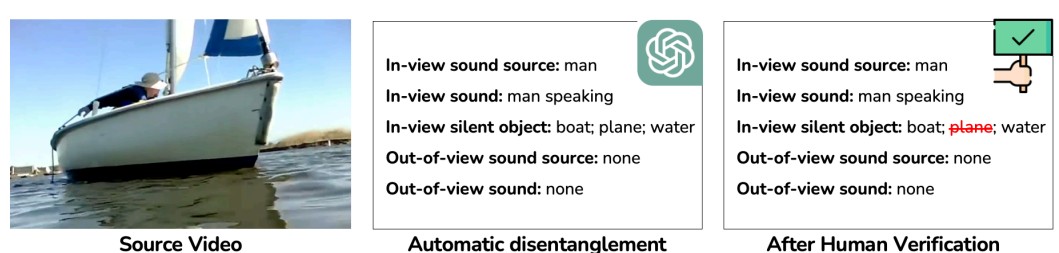

(a) A sample of error correction for Stage 1 outputs

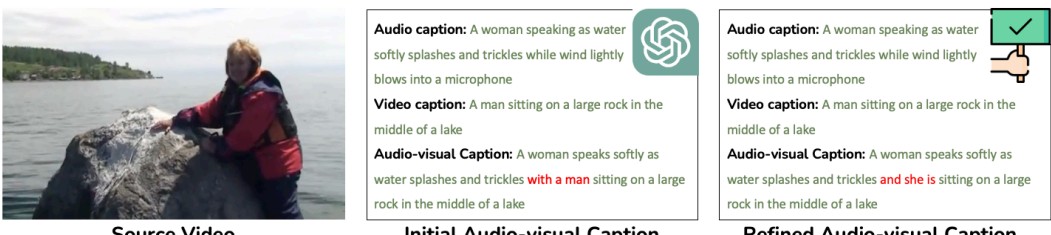

(b) A sample of error correction for Stage 2 outputs

**Figure S6: Samples of model-based annotation error correction in the data construction pipeline.** (a) Although the automatic disentanglement in Stage 1 may include noisy annotations due to reliance on off-the-shelf models, we observe that human annotators effectively identify and correct these inaccuracies. Similarly, (b) The initial audio-visual captions in Stage 2 may inaccurately describe the given video, as ChatGPT utilizes unaligned audio-only and video-only captions that may contain errors for generating audio-visual captions. However, human verification plays a crucial role in recognizing and correcting these errors.

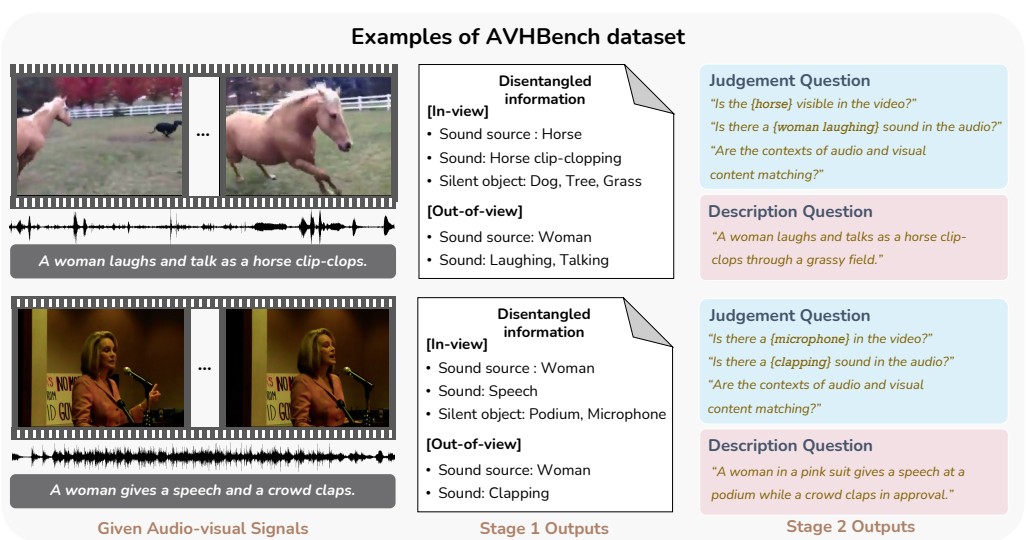

**Figure S7: Samples from AVHBench dataset.** Given the audio-visual input signal, Stage 1 outputs the disentangled audio and visual information. Stage 2 then utilizes the outputs from Stage 1 and generate QnA pairs for judgment tasks and audio-visual caption for description task. Note that this figure shows the examples of the `AVHBench` dataset which is the final product after automatic pipeline and human verification.

Moreover, similar errors can occur in Stage 2, when ChatGPT uses audio-only and video-only captions to generate an audio-visual caption. Since captions from each modality could be misaligned, the language model (ChatGPT) may not always accurately capture their correspondence, leading to merging these misaligned audio-visual captions. For example, as shown in Fig. S6-(b), although the video caption incorrectly describes a man sitting on the rock instead of a woman, the initial audio-visual caption generated by ChatGPT merely merges the two captions from different modalities, regardless of the misalignment, resulting in an incorrect audio-visual caption. Nevertheless, during

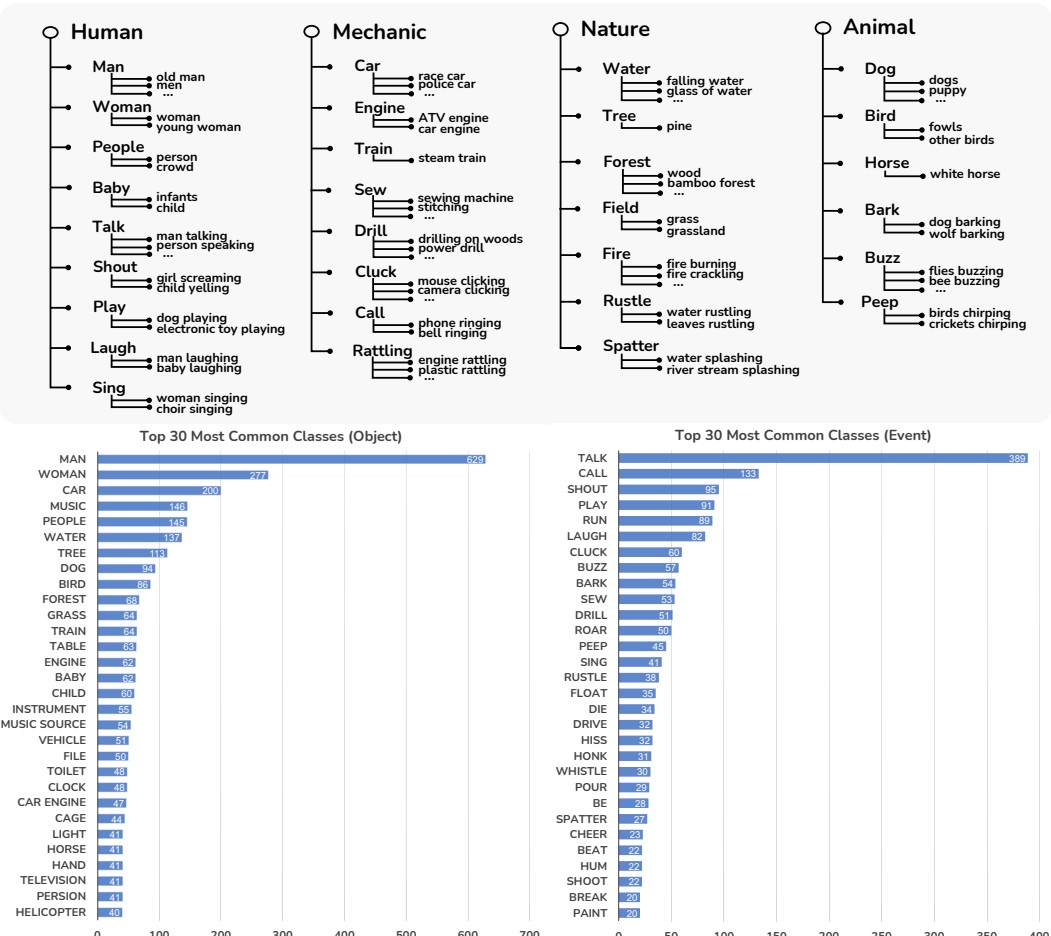

**Figure S8: Detailed dataset statistics.** [Top] is an ontology of word categories, and [Bottom] shows the bar graph of the most frequently appearing words in our benchmark.

human verification, the error is identified, and the caption is modified from "with a man" to "and she is," ensuring the refined audio-visual caption accurately describes the video.

## D.6    SAMPLE OUTPUTS OF EACH STAGE

Figure. S7 shows output samples from both Stage 1 and Stage 2 after automatic pipeline and human verification. Given the input audio-visual signals, such as audio (or audio-visual) captions and visual taggings, Stage 1 outputs the disentangled audio and visual information. Using this disentangled information, Stage 2 outputs QnA pairs for judgment tasks and audio-visual captions for the description task.

## D.7    DATASET STATISTIC

As the source video datasets (Kim et al., 2019; Chen et al., 2023b) do not provide class labels for each video, constructing class-level labels for our benchmark is infeasible. Therefore, we categorize the words used in the QnAs using WordNet (Miller, 1995) and visualize the ontology of the most frequently appearing words based on the top-level classes defined in AudioCaps (Kim et al., 2019) (Fig. S8-[top]). We also visualize a bar graph for the top 30 frequently used word categories (Fig. S8-[down]). As illustrated, the dataset encompasses diverse in-the-wild scenarios related to humans, mechanics, nature, and animals. As mentioned in the limitations section of the main text, expanding the diversity and complexity to cover more scenarios is a promising future direction.

# E    DETAILS ON THE EVALUATION SETTINGS

Our proposed benchmark, `AVHBench`, is evaluated on six different recent audio-visual LLMs (Panagopoulou et al., 2023; Han et al., 2023; Zhang et al., 2023a; Zhao et al., 2023c; Su et al., 2023; Han et al., 2024) capable of processing both auditory and visual information simultaneously. In this section, we introduce a parsing strategy for the non-binary model outputs (*i.e.*, neither "Yes" nor "No") in the judgment tasks, to quantitatively evaluate the models. Additionally, we provide further details of the baseline models that we used for the evaluations.

## E.1    ANSWER PARSING STRATEGY

To measure the quantitative evaluation metrics (*e.g.*, accuracy), we should properly parse the binary "Yes" or "No" answers from the generated outputs of each baseline model. For example, we parse "Yes" from an answer "Yes, the {`object/event+object`} is visible in the video." Specifically, a model might generate indefinitive answers which do not contain both "Yes" and "No", such as "I'm sorry, I cannot see any {`object/event+object`} in the video you provided." Thus, to transcribe them into the definitive answers, we carefully consider and classify all possible answer cases of generated outputs in a manual way. If the model fails to answer a definitive "Yes" or "No" answer over five trials despite careful consideration, we consider it as a wrong answer.

## E.2    BENCHMARKED MODELS

**X-InstructBlip.** X-InstructBlip (Panagopoulou et al., 2023) introduces a simple, yet effective cross-modality framework based on frozen LLMs that integrates various modalities without the need for modality-specific customization. This model employs the Q-Former (Li et al., 2023a) module, which is initialized with image-text pretrained weights from BLIP-2 (Li et al., 2023a). It then fine-tunes the Q-Former on an unimodal dataset to map inputs from separate modality spaces to the frozen LLM. Both the LLM and the encoders remain frozen, while only the Q-Former is trained to link the encoder and the LLM. X-InstructBlip utilizes the open-sourced Vicuna-7B (Chiang et al., 2023) as the LLM, and adopts ViT-G (Fang et al., 2023) for the video encoder, and BEATs (Chen et al., 2022) as the audio encoder. Input audios are processed through mono conversion and filterbank preprocessing, followed by normalization across two 5-second frames. Input videos are uniformly sampled to 5 frames and resized to $224 \times 224$ resolution with random cropping and normalization.

**ImageBind-LLM.** ImageBind-LLM (Han et al., 2023) leverages the vision-language data to align the joint embedding space of ImageBind (Girdhar et al., 2023) with LLaMA-7B (Touvron et al., 2023a). This model proposes a bind network with an attention-free zero-initialized injection method, which transforms the visual feature and directly adds it with every word token at all transformer layers of LLaMA. After the alignment, they fine-tune ImageBind-LLM utilizing high-quality visual instruction data from MiniGPT-4 (Chen et al., 2023a), to derive the multimodal instruction-following capacity. Furthermore, this model proposes cache-enhanced inference, which regards the ImageBind-encoded feature as the query and retrieves the top-$k$ similar visual keys from the cache model. This enhances the representation quality of other modalities, which may mitigate their semantic gaps with the images.

**Video-LLaMA.** Video-LLaMA (Zhang et al., 2023a) empowers frozen LLMs with the capability of comprehending both visual and auditory cues in videos. This model incorporates two branches: the vision-language branch and audio-language branch, to transform video frames and audio signals into query representations compatible with the textual input space of LLMs. Video-LLaMA trains each branch separately, with the audio-language branch only trained on visual-text data, following a similar data and process as the vision branch. This training is possible by the shared embedding space provided by ImageBind (Girdhar et al., 2023), which endows the audio interface with the ability to understand audio signals during inference. For evaluation, we employ Video-LLaMA which consists of pre-trained BLIP-2 (Li et al., 2023a), Vicuna-7B (Chiang et al., 2023), and ImageBind-huge.

**ChatBridge.** ChatBridge (Zhao et al., 2023c) employs open-source Vicuna-13B (Chiang et al., 2023) as the LLM, ViT-G as the vision encoder to encode images and videos, and BEAT (Chen et al., 2022) as the audio encoder to encode audio. Specifically, this model introduces a perceiver module composed of a transformer decoder to bridge the modal-specific encoders and the LLM. ChatBridge uses a shared perceiver for all modalities while each modality has its independent learnable query

tokens. Due to limited computation resources, only the perceivers and their learnable query tokens are trained while keeping the encoders and LLM frozen during the whole training process.

**PandaGPT.** PandaGPT (Su et al., 2023) connects the representation space of the multimodal encoders from ImageBind (Girdhar et al., 2023) and LLM from Vicuna (Chiang et al., 2023) via a linear projection layer. To align the feature space of the multimodal encoders and LLM, they train the model using image-text paired instruction-following data from (Liu et al., 2023b; Chen et al., 2023a) with additional LoRA (Hu et al., 2022) weight optimization. This approach involves aggregating information from diverse inputs (*e.g.*, video and audio) through a straightforward weighted sum of modality features within the feature space. We employ PandaGPT comprising Vicuna-13B and ImageBind-huge.

**OneLLM.** OneLLM (Han et al., 2024) is a multimodal LLM that aligns eight different modalities to a unified language representation. This integration is facilitated through a universal projection module that adapts individual modality encoders to a shared language model space, leveraging the advantages of dynamic routing to enhance modality fusion. Furthermore, the training involves a progressive alignment of modalities to language models, offering advancements over previous models that relied on separate modality encoders. It also includes a large-scale multimodal instruction dataset with two million items to train and refine its capabilities.

## F   TRAINING DETAILS ON VIDEO-LLAMA AUDIO ALIGNMENT AND LORA FINE-TUNING

In the experiments described in the main paper, we have improved the existing audio-visual LLMs against cross-modal hallucination through Low-Rank Adaptation (LoRA) (Hu et al., 2022) fine-tuning and enhanced audio alignment. Specifically, we utilize Video-LLaMA for fine-tuning and audio alignment experiments, which incorporates the pre-trained models such as BLIP-2(Li et al., 2023a), Vicuna-7B (Chiang et al., 2023), and Imagebind-huge (Girdhar et al., 2023).

**Audio alignment.**   To enhance the alignment between audio features and LLM, we pre-train Q-Former (Li et al., 2023a) and a linear layer that link the frozen audio encoder and LLM, while keeping the visual-text branch and the LLM fixed. We train the audio-text branch of Video-LLaMA using total 173K audio-text pairs collected from AudioCaps (Kim et al., 2019) (50K), Clotho (Drossos et al., 2020) (14K), Audioset-SL (Hershey et al., 2021) (108K) and SoundBible (Mei et al., 2023) (1K). The training objective is to minimize the next token prediction loss (*i.e.*, cross-entropy loss) over the textual responses, enhancing the model's capability to understand and map audio data to textual representations. We utilize 4 A6000 (48GB) for distributed training with batch size 32 per device and an initial learning rate (3e-5) and weight decay (0.05) for 1 epoch. We leverage mixed precision that uses fp16 for multiplication and fp32 for addition.

**LoRA fine-tuning.**   We fine-tune Video-LLaMA using the training split of our `AVHBench` dataset, which consists of 10,327 videos with 87,624 QnA pairs. The training dataset is also constructed through the automatic pipeline outlined in Section. 3.2 of the main paper, without any human verification process. We apply LoRA to the query, key, value, and output projections of Vicuna-7B, while the other parameters are frozen. We set the rank and alpha value of LoRA to 16 and 32, respectively. The training objective remains the same as the audio alignment stage. We utilize 4 NVIDIA A6000 (48GB) for distributed training with batch size 8 per device and an initial learning rate (3e-5) and weight decay (0.05) for 1 epoch. We also leverage mixed precision that uses fp16 for multiplication and fp32 for addition.

## G   GPT4 PROMPTS FOR GAVIE METRIC

In addition to the existing captioning metrics for evaluating the Audio-visual Captioning task, as described in the main paper, we employ the GPT4-Assisted Visual Instruction Evaluation (GAVIE) (Liu et al., 2023a) metric. GAVIE uses GPT4 to assess the output of the LLM from two perspectives: relevancy (GAVIE-R), which measures how well the output follows the given instruction, and accuracy (GAVIE-A), which evaluates the hallucination in the generated response. Since the GAVIE metric was originally proposed for evaluating captions generated from visual-only inputs, we slightly modify the GPT4 prompt to suit the Audio-visual Captioning task evaluation. Furthermore, given the

**Table S5: Prompt for measuring GAVIE metric.** The provided prompt is fed into GPT4 to evaluate the relevancy and accuracy of the generated audio-visual caption compared to the ground-truth. The answers from 1 to 5 are example audio-visual captions generated by different models to be evaluated.

---

Given a video with following audio-visual caption:
As a duck quacks, the sun peeks through the clouds in a dark forest.
Here is the instruction for the video: Describe what you see and hear.

**answer1**: ducks quacking
**answer2**: A close-up shot of a tree with a dead branch and a bird perched on it, with a dark sky and a red sunset in the background.
**answer3**: As the video begins, we see a beautiful sunset over a forest. The sky is a mix of pink and orange hues, and the trees are bare. The wind is blowing gently, causing the leaves to rustle. We then see a close-up of a tree with no leaves, and the sun is setting behind it. The sky is a deep shade of purple, and the tree is silhouetted against it. Next, we see a tree with no leaves against a blue sky. The sun is setting, and the sky is a mix of pink and orange. The tree is bare, and the wind is blowing gently. Finally, we see a tree with no leaves against a pink sky. The sun is setting, and the sky is a mix of pink and orange. The tree is bare, and the wind is blowing gently. Throughout the video, we hear the sound of the wind blowing through the trees and the rustling of leaves.
**answer4**: The image shows a forest with trees and bushes, and the sound of leaves rustling can be heard in the background.
**answer5**: A man is walking in the woods and a duck is quacking in the background

Suppose you are a smart teacher, after looking at the audio-visual caption above, please score the above answers(0-10) according to the following criteria:
1: whether the response directly follows the instruction.
2: whether the response is accurate concerning the audio-visual caption.

relevancy:
score of answer1:
reason:
score of answer2:
reason:
score of answer3:
reason:
score of answer4:
reason:
score of answer5:
reason:

accuracy:
score of answer1:
reason:
score of answer2:
reason:
score of answer3:
reason:
score of answer4:
reason:
score of answer5:
reason:

---

relative simplicity of our task instruction, we only report GAVIE-A in our main paper. The complete prompt used for measuring the GAVIE metric is detailed in Table. S5.

**Leaderboard**

**Audio-driven Video Hallucination**

| Rank | Model | Acc. (↑) | Precision (↑) | Recall (↑) | F1 (↑) | Yes (%) |
|------|-------|----------|---------------|------------|--------|---------|
| 🥇 1st | Gemini-Flash | 83.3 | 85.7 | 81.0 | 83.7 | 47.3 |
| 🥈 2nd | Video-SALMONN | 78.1 | 74.9 | 84.5 | 79.4 | 56.4 |
| 🥉 3rd | Video-LLaMA2 | 75.2 | 73.6 | 78.7 | 76.1 | 53.6 |
| 4th | PandaGPT | 58.5 | 55.3 | 91.1 | 68.8 | 82.3 |
| 5th | OneLLM | 53.7 | 58.6 | 64.8 | 49.8 | 63.1 |
| 6th | ChatBridge | 52.9 | 70.9 | 52.9 | 48.9 | 77.6 |
| 7th | ImageBind-LLM | 50.3 | 50.2 | 87.1 | 63.7 | 86.7 |
| 8th | Video-LLaMA | 50.1 | 50.1 | 100 | 66.7 | 99.9 |
| 9th | X-InstrcutBLIP | 18.1 | 16.0 | 15.0 | 15.5 | 46.9 |

**Video-driven Audio Hallucination**

| Rank | Model | Acc. (↑) | Precision (↑) | Recall (↑) | F1 (↑) | Yes (%) |
|------|-------|----------|---------------|------------|--------|---------|
| 🥇 1st | Video-LLaMA2 | 74.2 | 69.4 | 86.6 | 77.0 | 62.4 |
| 🥈 2nd | Video-SALMONN | 65.2 | 62.3 | 76.9 | 68.8 | 61.7 |
| 🥉 3rd | Gemini-Flash | 63.0 | 57.9 | 94.7 | 71.9 | 81.7 |
| 4th | PandaGPT | 61.3 | 57.4 | 86.6 | 69.1 | 75.5 |
| 5th | Video-LLaMA | 50.2 | 50.2 | 100 | 66.9 | 100 |
| 6th | ImageBind-LLM | 50.0 | 50.0 | 99.3 | 66.5 | 99.3 |
| 7th | OneLLM | 44.3 | 50.2 | 39.4 | 49.8 | 55.0 |
| 8th | ChatBridge | 32.8 | 60.0 | 32.8 | 39.8 | 14.8 |
| 9th | X-InstrcutBLIP | 16.3 | 14.5 | 38.5 | 21.1 | 77.0 |

**Figure S9: Dataset maintenance.** We plan to maintain a benchmark leaderboard webpage to facilitate ongoing evaluation of audio-visual LLMs.

# H MAINTENANCE

To further support ongoing research on audio-visual LLMs, we plan to maintain a benchmark leaderboard webpage as shown in Fig. S9. The leaderboard will list models we have already evaluated and will be continuously updated as new audio-visual LLMs are tested on our benchmark.

