# OpenReview forum: "AVHBench: A Cross-Modal Hallucination Benchmark for Audio-Visual Large Language Models"
_ICLR.cc/2025/Conference — ICLR 2025 Poster_

### Official Review · Reviewer_huna · 2024-10-21

**Soundness:** 3
**Presentation:** 2
**Contribution:** 3
**Rating:** 6
**Confidence:** 5

**Summary:**

This paper proposed AVHBench, a benchmark to assess cross-modal hallucinations in audio-visual LLMs. The benchmark is built on existing datasets VALOR and AudioCaps, and contains about 6k QnA pairs and 1k audio-visual captions across four cross-modal tasks, including audio-driven video hallucination, video-driven audio hallucination, audio-visual matching and audio-visual captioning. The authors design a semi-automatic pipeline for data anntation. Several open-source audio-visual LLMs are evaluated on AVHBench, and the results show that most existing audio-visual LLMs suffer from cross-modal hallucinations. To alleviate this problem, the authors further enhances Video-LLaMA through audio feature alignment and LoRA fine-tuning, proving that audio-visual hallucinations might come from insufficient training on paired audio-visual data.

**Strengths:**

- This paper proposed a audio-visual hallucination benchmark mainly focusing on cross-modal hallucination evaluation, which is an aspect that has received little attention.

- Some tasks in the benchmark provide new perspectives on the study of multi-modal hallucinations, like *Audio-driven Video Hallucination* and *Video-driven Audio Hallucination*. The inability of the model to distinguish between information from audio or video may be a vital reason that causes multi-modal hallucination.

- The paper is well-written, clear and easy to understand.

**Weaknesses:**

- The benchmark uses existing datasets VALOR and AudioCaps, which may introduce biases inherent to those datasets, potentially affecting the generalizability and validity of the evaluation.

- In this benchmark, for human speech, the authors seem to have only considered the event of "one person is speaking," instead of taking into account the content of what is being said. The content of speech contains a wealth of information and is very likely to contribute to the hallucinations of audio-visual LLMs. However, the paper seems to overlook this scenario.

    For example, consider a scenario where a person in a video is saying to himself "Yesterday I heard a dog barking", but neither the dog nor the barking sound appears in the video or audio. Models are prone to hallucinations in this scenario.

- It is recommended to evaluate more recent audio-visual LLMs, like Video-LLaMA 2, or video-SALMONN.

- Since the video is so rich in content, long text is required to descirbe the video completely. However, for the "audio-visual captioning" task, only a short caption is provided as the groundtruth. This suggests that the groundtruth caption is likely to contain only the important information in the video and omit the secondary information. However, it is still possible for the model to describe something that is present in the video but is not hallucination. That's why I'm concerned about the correctness of the "audio-visual captioning" task of AVHBench.

**Questions:**

- After LoRA fine-tuning, the results of the model on AVHBench become significantly better. Does this suggest that the model may just not be able to do the judgement questions, or that it hasn't seen the mismatched audio/video and therefore performs poorly on that test set, rather than the model having a large number of hallucinations?

- Do the authors consider providing Gemini's results on the benchmark?

---

> ### Author Response · Authors · 2024-11-22
> **Author response (1/3)**
>
> We thank reviewer huna for recognizing our work as providing a new perspective on multimodal hallucination, and for delivering a well-written and clear paper, along with constructive feedback. We address the concerns raised by reviewer huna below, and encourage the reviewer huna to check the newly updated revision PDF that includes the requested content. The red-colored texts indicate the newly added part in the revision PDF.
>
> **Generalizability and validity of the evaluation**
> >We wish to clarify that our primary goal is to serve as a cornerstone research for evaluating cross-modal hallucinations, i.e., assessing how one modality might affect the perception of another. While our dataset, built upon VALOR and AudioCaps, may not encompass all real-world scenarios, it does cover a diverse range of in-the-wild scenarios related to human actions, mechanics, nature, and animal-related events, as illustrated in Figure S8 in the Appendix of the initial submission. As the original purposes and generalities of each existing dataset have already been acknowledged by their own, we believe that their union coverage in our work is still valid to evaluate such hallucinations in audio-visual LLMs. Nevertheless, we concur that our dataset may not cover all scenarios and including more scenarios and datasets would make this benchmark more comprehensive. This point has already been discussed in "Section 5. Conclusion and Discussion" of our initial submission, and we leave more extensions for future work.
>
> **Content of speech may introduce hallucination**
> >We agree that speech (linguistic content) may induce hallucinations and recognize that addressing this could be an entirely new impactful direction for research, making the dataset more comprehensive. One approach could involve integrating speech content into our dataset construction pipeline through Automatic Speech Recognition (ASR). This addition would help explore how speech influences visual perception hallucinations, but it becomes a completely different research. We have newly added Section 5 Limitation and future work in the revision PDF, and included this discussion in Line 485-504.
> We’d like to highlight that, as acknowledged by all the reviewers, our benchmark focuses on the scope that has barely been explored and is still valuable in the community. As mentioned in Sec. 3.2. in our initial submission, our benchmark focuses on environmental- or event-level videos and audios. While few studies dealing with both speech and environment sounds have recently begun, the vast majority of audio-video language models still rely on audio encoders only dealing with environmental audio, not speech. We’d like to ask the reviewer to recognize this contribution.
>
> **Evaluation on more recent audio-visual models**
> >As requested by reviewer huna, **we have conducted additional evaluations on our benchmark with 5 more recent models**; audio-visual LLMs, including Video-LLaMA2, Video-SALMONN, and Gemini, and unimodal LLMs, including LLaVA-Onevision, and Qwen2-audio. The results are summarized in the provided table and these new results and discussions are **newly updated in Appendix Section C. Additional quantitative results of the revision PDF.**
>
> >We observe that overall performance on the benchmark has improved with these newer models compared to those used in our main paper. **Interestingly, we find that overall model performance improves when using unimodal signals for evaluation (comparing results from Audio-driven Video Hallucination vs. Video Hallucination; Video-driven Audio Hallucination vs. Audio Hallucination).** Specifically, Video-LLaMA2 demonstrates notable improvements with unimodal signals compared to using both audio-visual signals, supporting discussions in Section 4.2 of the initial submission on how cross-modal signals can confuse models and evoke hallucinations.
> These findings highlight recent enhancements in models to address cross-modal hallucination and also demonstrate the effectiveness of our benchmark in evaluating these advancements.
>
> >Furthermore, **we plan to maintain a benchmark leaderboard webpage to facilitate ongoing evaluation of audio-visual LLMs.** We have **newly updated this captured photo of the leaderboard in Appendix Section H Maintenance of the revision PDF.**

---

> > ### Author Response · Authors · 2024-11-22
> > **Author response (2/3)**
> >
> > |  Model    | Audio-driven Video Hallucination |                  |                  |                  |                  | Video-driven Audio Hallucination |                  |                  |                  |                  |
> > |---------------------|----------------------------------|------------------|------------------|------------------|------------------|----------------------------------|------------------|------------------|------------------|------------------|
> > |             | Acc. (↑)                       | Precision (↑)    | Recall (↑)       | F1 (↑)           | Yes (%)          | Acc. (↑)                       | Precision (↑)    | Recall (↑)       | F1 (↑)           | Yes (%)          |
> > | Video-LLaMA         | 50.1                            | 50.1             | **100**              | 66.7             | 99.9             | 50.2                            | 50.2             | **100**              | 66.9             | 100              |
> > | Video-SALMONN       | 78.1                            | 74.9             | 84.5             | 79.4             | 56.4             | 65.2                            | 62.3             | 76.9             | 68.8             | 61.7             |
> > | Video-LLaMA2        | 75.2                            | 73.6             | 78.7             | 76.1             | 53.6             | **74.2**                            | **69.4**             | 86.6             | **77.0**             | 62.4             |
> > | Gemini-Flash        | **83.3**                            | **85.7**             | 81.0             | **83.7**             | 47.3             | 63.0                            | 57.9             | 94.7             | 71.9             | 81.7             |
> >
> >
> > | Model                      | Video Hallucination             |                  |                  |                  |                  | Audio Hallucination             |                  |                  |                  |                  |
> > |----------------------------|----------------------------------|------------------|------------------|------------------|------------------|----------------------------------|------------------|------------------|------------------|------------------|
> > | Model                      | Acc. (↑)                       | Precision (↑)    | Recall (↑)       | F1 (↑)           | Yes (%)          | Acc. (↑)                       | Precision (↑)    | Recall (↑)       | F1 (↑)           | Yes (%)          |
> > | Video-LLaMA                | 50.0                            | 50.0             | **100**              | 66.7             | **100**              | 58.4                            | 55.7             | 93.3             | 69.8             | 86.2             |
> > | Video-SALMONN              | 82.3                            | 83.9             | 79.9             | 81.9             | 47.6             | 66.5                            | 70.5             | 56.8             | 62.9             | 40.3             |
> > | Video-LLaMA2               | 80.7                            | 87.0             | 72.1             | 78.8             | 41.4             | **79.8**                            | **81.9**             | 76.4             | **79.1**             | 46.7             |
> > | Gemini-Flash               | **84.3**                            | 81.6             | 72.2             | 76.6             | 31.4             | 65.2                            | 59.6             | **94.3**             | 73.0             | 79.1             |
> > | PLLaVA (Unimodal)          | 81.5                            | 77.9             | 87.8             | **82.6**             | 56.3             | -                               | -                | -                | -                | -                |
> > | LLaVA-OneVision (Unimodal) | 84.0                            | **90.9**             | 75.5             | 82.5             | 41.5             | -                               | -                | -                | -                | -                |
> > | LTU (Unimodal)                       | -                               | -                | -                | -                | -                | 67.2                            | 63.9             | 84.2             | 72.7             | 68.3
> > | Qwen2-Audio (Unimodal)               | -                               | -                | -                | -                | -                | 74.4                               | 81.3               | 68.8               | 74.5                | 13.6                |            |

---

> > > ### Author Response · Authors · 2024-11-22
> > > **Author response (3/3)**
> > >
> > > **Correctness of the “audio-visual captioning task”**
> > > >We agree that if the video becomes longer and more complex, a longer audio-visual caption is required to capture all the information presented in the video. We have newly added Section 5 Limitation and future work in the revision PDF, and included this discussion in Lines 481-484. Nevertheless, we’d like to first note that we follow the existing video and audio benchmark formats of using the 10-second of short clips in our benchmark. Second, we believe that our audio-visual captioning in the benchmark sufficiently contains objects, actions, and sound events within the 10-second short clips. As mentioned in Table S2 in the Appendix of our initial submission, the audio-visual caption generation involves rich video captions produced by visual tags and human-annotated audio captions provided by the original source and are further qualified by human annotators. These richly given pieces of information, as well as the language capability of ChatGPT with few-shot samples and human verification, suffice to capture most of the content in the video, compared to the video-only or audio-only captions presented by the original datasets (VALOR and AudioCaps). In short, while the long video scenario is an interesting scope as suggested by the reviewer, our focus in this work is 10-second videos, and our captions are verified and reasonable enough to describe our target video scenarios.
> > >
> > >
> > > **[Questions] Effect of the LoRA fine-tuning**
> > > >We believe that LoRA fine-tuning induced the model to correctly attend to each signal given the question. As shown in the attention map in Figure S3 of the Appendix of our initial submission, the existing model attends to the opposite signal, for example, attending to the audio signal when the question is asked about the visibility of a certain object, or attending to the video signal when the question is about the sound event, which may cause the hallucinated output. Thus, fine-tuning with the audio-visual dataset allows the model to accurately attend to the correct modality, resulting in significant improvement in the hallucination benchmark.
> > >
> > > **[Questions] Providing Gemini’s result**
> > > >We have newly updated Gemini’s results in Appendix Section C. Additional quantitative results of the revision PDF (please refer to the provided table above). While Gemini performs well on Audio-driven Video Hallucination and Video Hallucination compared to other audio-visual LLMs, it exhibits relatively poor performance on audio perception and understanding tasks. Furthermore, using an unimodal signal enhances performance, indicating that the phenomenon of cross-modal hallucination also exists in this model.

---

> > > > ### Author Response · Authors · 2024-11-25
> > > >
> > > > Dear Reviewer huna,
> > > >
> > > > We sincerely appreciate reviewer huna’s time and effort in reviewing our submissions and providing valuable feedback. We believe all the comments and suggestions made by the reviewer huna have been addressed, including the evaluations of our benchmark on additional 5 different MLLMs. We welcome any further comments or feedback that may help to enhance our work. We'd be happy to further discuss.
> > > > Thank you.

---

> > > > > ### Comment · Reviewer_huna · 2024-11-25
> > > > > **Response to the authors**
> > > > >
> > > > > Thank you for the detailed response. The additional evaluations of more recent models have indeed enhanced the completeness of the paper. I recognize the novelty of this work; however, I still believe that the visual hallucinations caused by the speech content cannot be overlooked. Therefore, I will maintain my original score.

---

> ### Author Response · Authors · 2024-11-26
>
> We thank reviewer huna for their valuable feedback. However, we respectfully request that reviewer huna reconsider our work, which focuses distinctly on evaluating hallucinations in concurrent generic audio-visual signals. This research serves as a cornerstone for studies on audio-visual cross-modal hallucinations, with the potential to extend to speech in the future. Our main focus aligns with recent advancements in audio-visual LLMs that process generic audio inputs rather than speech (e.g., X-InstructBLIP, ImageBind-LLM, Video-LLaMA, Video-LLaMA2, ChatBridge, PandaGPT, OneLLM, LTU). Speech-integrating MLLMs, such as Video-Salmonn, have only recently been proposed. While we agree that speech is an important auditory signal, generic audio is also very crucial for evaluating the audio-visual understanding of MLLMs. Involving datasets like HowTo100M or VideoChat to address speech hallucination would indeed be valuable but represent a completely different research direction. We respectfully request reviewer huna reassess our work according to the scope of generic audio rather than introducing an out-of-scope context. Thank you.

---

> > ### Author Response · Authors · 2024-11-28
> >
> > Dear Reviewer huna,
> >
> > We would like to respectfully inquire whether there are any remaining concerns or aspects of our paper that are unclear and that we can address during the rebuttal period. As this period is meant to resolve issues raised during the review process, we kindly request that the reviewers provide us with more specific feedback or highlight any areas that may still require improvement, so that we can better address them.
> > If there are no further issues, we would greatly appreciate hearing whether the clarifications and revisions we have provided have sufficiently addressed reviewer huna’s concerns, and if reviewer huna would be willing to reassess our work.
> > Thank you.
> >
> > Best,
> > The Authors

---

### Official Review · Reviewer_7vLX · 2024-11-01

**Soundness:** 3
**Presentation:** 3
**Contribution:** 2
**Rating:** 5
**Confidence:** 4

**Summary:**

**Summary of this paper:**

This work proposes a cross-modal hallucination evaluation benchmark called AVHBench, which comprises four different tasks: audio-driven video hallucination, video-driven audio hallucination, audio-visual matching, and audio-visual captioning. Besides, the paper analyzes the presence of cross-modal hallucinations and investigating their potential causes using the proposed benchmark on six recent audio-visual LLMs.

**Strengths:**

1.	This paper introduces the first comprehensive benchmark specifically designed to evaluate the perception and comprehension capabilities of audio-visual LLMs.

2.	Authors include a clear organization of the related literature on multimodal large language models (MLLMs) and hallucinations in MLLMs.

3.	The figures in this paper are well-designed.

**Weakness:**

The datasets used in this paper are relatively limited in terms of scene diversity, which may hinder the benchmark's ability to evaluate the model's performance in a broader range of real-world scenarios.

**Comments, Suggestions And Typos:**

To provide a more comprehensive evaluation of this benchmark, it is recommended to increase the number of evaluation models and diversify the scenarios.

**Strengths:**

Refer to the summary

**Weaknesses:**

Refer to the summary

**Questions:**

Refer to the summary

---

> ### Author Response · Authors · 2024-11-22
> **Author response (1/3)**
>
> We thank reviewer 7vLX for recognizing our work as providing the first comprehensive benchmark for audio-visual LLMs, well-designed figures, and clear organization, along with constructive feedback. Here we address the concerns raised by reviewer 7vLX, and encourage the reviewer 7vLX to check the newly updated revision PDF that includes the requested content. The red colored texts indicate the newly added part in the revision PDF. We respectfully request the reviewer 7vLX to reassess our work and consider increasing the rating.
>
> **Increasing the number of evaluation models**
> >As requested by reviewer 7vLX, **we have conducted additional evaluations on our benchmark with 5 more recent models**; audio-visual LLMs, including Video-LLaMA2, Video-SALMONN, and Gemini, and unimodal LLMs, including LLaVA-Onevision, and Qwen2-audio. The results are summarized in the provided table and these new results and discussions are **newly updated in Appendix Section C. Additional quantitative results of the revision PDF.**
>
> >We observe that overall performance on the benchmark has improved with these newer models compared to those used in our main paper. **Interestingly, we find that overall model performance improves when using unimodal signals for evaluation (comparing results from Audio-driven Video Hallucination vs. Video Hallucination; Video-driven Audio Hallucination vs. Audio Hallucination).** Specifically, Video-LLaMA2 demonstrates notable improvements with unimodal signals compared to using both audio-visual signals, supporting discussions in Section 4.2 of the initial submission on how cross-modal signals can confuse models and evoke hallucinations.
> These findings highlight recent enhancements in models to address cross-modal hallucination and also demonstrate the effectiveness of our benchmark in evaluating these advancements.
>
> >Furthermore, **we plan to maintain a benchmark leaderboard webpage to facilitate ongoing evaluation of audio-visual LLMs.** We have **newly updated this captured photo of the leaderboard in Appendix Section H Maintenance of the revision PDF.**

---

> > ### Author Response · Authors · 2024-11-22
> > **Author response (2/3)**
> >
> > |  Model    | Audio-driven Video Hallucination |                  |                  |                  |                  | Video-driven Audio Hallucination |                  |                  |                  |                  |
> > |---------------------|----------------------------------|------------------|------------------|------------------|------------------|----------------------------------|------------------|------------------|------------------|------------------|
> > |             | Acc. (↑)                       | Precision (↑)    | Recall (↑)       | F1 (↑)           | Yes (%)          | Acc. (↑)                       | Precision (↑)    | Recall (↑)       | F1 (↑)           | Yes (%)          |
> > | Video-LLaMA         | 50.1                            | 50.1             | **100**              | 66.7             | 99.9             | 50.2                            | 50.2             | **100**              | 66.9             | 100              |
> > | Video-SALMONN       | 78.1                            | 74.9             | 84.5             | 79.4             | 56.4             | 65.2                            | 62.3             | 76.9             | 68.8             | 61.7             |
> > | Video-LLaMA2        | 75.2                            | 73.6             | 78.7             | 76.1             | 53.6             | **74.2**                            | **69.4**             | 86.6             | **77.0**             | 62.4             |
> > | Gemini-Flash        | **83.3**                            | **85.7**             | 81.0             | **83.7**             | 47.3             | 63.0                            | 57.9             | 94.7             | 71.9             | 81.7             |
> >
> >
> > | Model                      | Video Hallucination             |                  |                  |                  |                  | Audio Hallucination             |                  |                  |                  |                  |
> > |----------------------------|----------------------------------|------------------|------------------|------------------|------------------|----------------------------------|------------------|------------------|------------------|------------------|
> > | Model                      | Acc. (↑)                       | Precision (↑)    | Recall (↑)       | F1 (↑)           | Yes (%)          | Acc. (↑)                       | Precision (↑)    | Recall (↑)       | F1 (↑)           | Yes (%)          |
> > | Video-LLaMA                | 50.0                            | 50.0             | **100**              | 66.7             | **100**              | 58.4                            | 55.7             | 93.3             | 69.8             | 86.2             |
> > | Video-SALMONN              | 82.3                            | 83.9             | 79.9             | 81.9             | 47.6             | 66.5                            | 70.5             | 56.8             | 62.9             | 40.3             |
> > | Video-LLaMA2               | 80.7                            | 87.0             | 72.1             | 78.8             | 41.4             | **79.8**                            | **81.9**             | 76.4             | **79.1**             | 46.7             |
> > | Gemini-Flash               | **84.3**                            | 81.6             | 72.2             | 76.6             | 31.4             | 65.2                            | 59.6             | **94.3**             | 73.0             | 79.1             |
> > | PLLaVA (Unimodal)          | 81.5                            | 77.9             | 87.8             | **82.6**             | 56.3             | -                               | -                | -                | -                | -                |
> > | LLaVA-OneVision (Unimodal) | 84.0                            | **90.9**             | 75.5             | 82.5             | 41.5             | -                               | -                | -                | -                | -                |
> > | LTU (Unimodal)                       | -                               | -                | -                | -                | -                | 67.2                            | 63.9             | 84.2             | 72.7             | 68.3
> > | Qwen2-Audio (Unimodal)               | -                               | -                | -                | -                | -                | 74.4                               | 81.3               | 68.8               | 74.5                | 13.6                |            |

---

> ### Author Response · Authors · 2024-11-22
> **Author response (3/3)**
>
> **Limitation in evaluating a broader range of real-world scenarios**
> >Although our dataset does not encompass every real-world scenario (we have acknowledged limitations and have already described it in "Section 5. Conclusion and Discussion"), we believe the diversity is reasonable for our dataset as an important cornerstone research for evaluating cross-modal hallucinations in audio-visual LLMs. Our dataset includes a broad range of in-the-wild situations involving human actions, mechanics, nature, and animal-related events, as shown in Figure S8 in the Appendix. This is a virtue of our benchmark constructed by integrating multiple existing datasets; thus, our benchmark is much more diverse than the test subsets of each individual dataset. Moreover, as mentioned in Section D.7 of the Appendix, we are expanding the dataset after the main submission to incorporate even more diverse scenarios.
>
> >Furthermore, please refer to the response to reviewer MmyJ, where we have added discussions on diverse scenarios, such as ensuring visuals or audio feature objects that are silent or absent, handling ambient sounds like wind, managing multiple objects of the same type with different statuses, and addressing background music. These points highlight that our dataset already captures a wide variety of meaningful and challenging aspects.
>
> >To avoid confusion and better address concerns, it would be helpful if you could provide specific suggestions on the aspects that should be included to diversify the scenarios.

---

> > ### Author Response · Authors · 2024-11-25
> >
> > Dear Reviewer 7vLX,
> >
> > We sincerely appreciate reviewer 7vLX’s time and effort in reviewing our submissions and providing valuable feedback. We believe all the comments and experiments suggested by the reviewer 7vLX have been addressed, including the evaluations of our benchmark on additional 5 different MLLMs. If reviewer 7vLX has no further concerns, we respectfully request the reviewer to reassess our work and consider increasing the rating. Thank you.

---

> > > ### Author Response · Authors · 2024-11-26
> > >
> > > Dear Reviewer 7vLX,
> > >
> > > We sincerely request the reviewer to review the rebuttal that we had provided. We believe we have addressed all the comments and experiments raised by the reviewer through our detailed responses and clarifications. We would appreciate your prompt attention before the discussion phase concludes. A thorough reassessment will help to ensure a fair evaluation of our work. Thank you for your immediate attention to this matter.
> > >
> > > Best regards, The Authors

---

> > > > ### Author Response · Authors · 2024-11-28
> > > >
> > > > Dear Reviewer 7vLX,
> > > >
> > > > We would like to respectfully inquire whether there are any remaining concerns or aspects of our paper that are unclear and that we can address during the rebuttal period. As this period is meant to resolve issues raised during the review process, we kindly request that the reviewers provide us with more specific feedback or highlight any areas that may still require improvement, so that we can better address them.
> > > > If there are no further issues, we would greatly appreciate hearing whether the clarifications and revisions we have provided have sufficiently addressed reviewer 7vLX’s concerns, and if reviewer 7vLX would be willing to reassess our work.
> > > > Thank you.
> > > >
> > > > Best,
> > > > The Authors

---

> > > > > ### Author Response · Authors · 2024-12-02
> > > > >
> > > > > Dear Reviewer 7vLX,
> > > > >
> > > > > We sincerely ask Reviewer 7vLX if there are any remaining concerns. In our rebuttal, we have addressed the issues raised by Reviewer 7vLX by conducting additional evaluations on five more models (Video-SALMONN, Gemini, Qwen-Audio, Video-LLaMA2, and LLaVA-OneVision). Additionally, we have provided clarifications regarding the dataset coverage. Since this is the last day of the discussion period, if there are no further issues, we would greatly appreciate reassessing our work and considering increasing the rating. Thank you.
> > > > >
> > > > > Best, The Authors

---

### Official Review · Reviewer_FJwk · 2024-11-02

**Soundness:** 2
**Presentation:** 3
**Contribution:** 3
**Rating:** 6
**Confidence:** 3

**Summary:**

The paper introduces AVHBench, a novel benchmark designed to evaluate cross-modal hallucinations in audio-visual large language models (LLMs). Addressing a critical gap, AVHBench tests models on their ability to handle complex interactions between audio and visual cues without generating erroneous outputs, known as cross-modal hallucinations. It includes four tasks: audio-driven video hallucination, video-driven audio hallucination, audio-visual matching, and captioning. Using a semi-automated pipeline for dataset curation, AVHBench facilitates robust assessment and enhancement of model accuracy in handling multimodal inputs.

**Strengths:**

1. This paper is well-written.
2. The motivation for proposing benchmark for audio-visual hallucination is clear.
3. The tasks proposed by the benchmark are valuable, and the semi-automatic solutions designed are also reasonable.

**Weaknesses:**

As a benchmark paper, more evaluation models are needed. For example, the recent video-salmonn[1], advanced Gemini[2], and unimodal models like Qwen-audio[3,4] for audio, llava-onevision[5] for vision.

1. Sun, Guangzhi, et al. "video-SALMONN: Speech-enhanced audio-visual large language models." arXiv preprint arXiv:2406.15704 (2024).
2. Team, Gemini, et al. "Gemini: a family of highly capable multimodal models." arXiv preprint arXiv:2312.11805 (2023).
3. Chu, Yunfei, et al. "Qwen-audio: Advancing universal audio understanding via unified large-scale audio-language models." arXiv preprint arXiv:2311.07919 (2023).
4. Chu, Yunfei, et al. "Qwen2-audio technical report." arXiv preprint arXiv:2407.10759 (2024).
5. Li B, Zhang Y, Guo D, et al. Llava-onevision: Easy visual task transfer[J]. arXiv preprint arXiv:2408.03326, 2024.

**Questions:**

see weaknesses.

---

> ### Author Response · Authors · 2024-11-22
> **Author response (1/2)**
>
> We thank reviewer FJwk for recognizing our work as proposing a valuable benchmark, reasonable semi-automatic pipeline, and well-written with clear motivation, along with constructive feedback. We address the concerns raised by reviewer FJwk, and encourage the reviewer FJwk to check the newly updated revision PDF that includes the requested content.
>
> **More evaluation on more recent audio-visual models**
> >As requested by reviewer FJwk, **we have conducted additional evaluations on our benchmark with 5 more recent models**; audio-visual LLMs, including Video-LLaMA2, Video-SALMONN, and Gemini, and unimodal LLMs, including LLaVA-Onevision, and Qwen2-audio. The results are summarized in the provided table and these new results and discussions are **newly updated in Appendix Section C. Additional quantitative results of the revision PDF.**
>
> >We observe that overall performance on the benchmark has improved with these newer models compared to those used in our main paper. **Interestingly, we find that overall model performance improves when using unimodal signals for evaluation (comparing results from Audio-driven Video Hallucination vs. Video Hallucination; Video-driven Audio Hallucination vs. Audio Hallucination).** Specifically, Video-LLaMA2 demonstrates notable improvements with unimodal signals compared to using both audio-visual signals, supporting discussions in Section 4.2 of the initial submission on how cross-modal signals can confuse models and evoke hallucinations.
> These findings highlight recent enhancements in models to address cross-modal hallucination and also demonstrate the effectiveness of our benchmark in evaluating these advancements.
>
> >Furthermore, **we plan to maintain a benchmark leaderboard webpage to facilitate ongoing evaluation of audio-visual LLMs.** We have **newly updated this captured photo of the leaderboard in Appendix Section H Maintenance of the revision PDF.**

---

> > ### Author Response · Authors · 2024-11-22
> > **Author response (2/2)**
> >
> > |  Model    | Audio-driven Video Hallucination |                  |                  |                  |                  | Video-driven Audio Hallucination |                  |                  |                  |                  |
> > |---------------------|----------------------------------|------------------|------------------|------------------|------------------|----------------------------------|------------------|------------------|------------------|------------------|
> > |             | Acc. (↑)                       | Precision (↑)    | Recall (↑)       | F1 (↑)           | Yes (%)          | Acc. (↑)                       | Precision (↑)    | Recall (↑)       | F1 (↑)           | Yes (%)          |
> > | Video-LLaMA         | 50.1                            | 50.1             | **100**              | 66.7             | 99.9             | 50.2                            | 50.2             | **100**              | 66.9             | 100              |
> > | Video-SALMONN       | 78.1                            | 74.9             | 84.5             | 79.4             | 56.4             | 65.2                            | 62.3             | 76.9             | 68.8             | 61.7             |
> > | Video-LLaMA2        | 75.2                            | 73.6             | 78.7             | 76.1             | 53.6             | **74.2**                            | **69.4**             | 86.6             | **77.0**             | 62.4             |
> > | Gemini-Flash        | **83.3**                            | **85.7**             | 81.0             | **83.7**             | 47.3             | 63.0                            | 57.9             | 94.7             | 71.9             | 81.7             |
> >
> >
> > | Model                      | Video Hallucination             |                  |                  |                  |                  | Audio Hallucination             |                  |                  |                  |                  |
> > |----------------------------|----------------------------------|------------------|------------------|------------------|------------------|----------------------------------|------------------|------------------|------------------|------------------|
> > | Model                      | Acc. (↑)                       | Precision (↑)    | Recall (↑)       | F1 (↑)           | Yes (%)          | Acc. (↑)                       | Precision (↑)    | Recall (↑)       | F1 (↑)           | Yes (%)          |
> > | Video-LLaMA                | 50.0                            | 50.0             | **100**              | 66.7             | **100**              | 58.4                            | 55.7             | 93.3             | 69.8             | 86.2             |
> > | Video-SALMONN              | 82.3                            | 83.9             | 79.9             | 81.9             | 47.6             | 66.5                            | 70.5             | 56.8             | 62.9             | 40.3             |
> > | Video-LLaMA2               | 80.7                            | 87.0             | 72.1             | 78.8             | 41.4             | **79.8**                            | **81.9**             | 76.4             | **79.1**             | 46.7             |
> > | Gemini-Flash               | **84.3**                            | 81.6             | 72.2             | 76.6             | 31.4             | 65.2                            | 59.6             | **94.3**             | 73.0             | 79.1             |
> > | PLLaVA (Unimodal)          | 81.5                            | 77.9             | 87.8             | **82.6**             | 56.3             | -                               | -                | -                | -                | -                |
> > | LLaVA-OneVision (Unimodal) | 84.0                            | **90.9**             | 75.5             | 82.5             | 41.5             | -                               | -                | -                | -                | -                |
> > | LTU (Unimodal)                       | -                               | -                | -                | -                | -                | 67.2                            | 63.9             | 84.2             | 72.7             | 68.3
> > | Qwen2-Audio (Unimodal)               | -                               | -                | -                | -                | -                | 74.4                               | 81.3               | 68.8               | 74.5                | 13.6                |            |

---

> > > ### Author Response · Authors · 2024-11-25
> > >
> > > Dear Reviewer FJwk,
> > >
> > > We sincerely appreciate reviewer FJwk’s time and effort in reviewing our submissions and providing valuable feedback. We believe that we have resolved the concern by conducting evaluations of our benchmark on 5 different models, including 3 audio-visual LLM models, and 2 uni-modal LLM models. If reviewer FJwk has no further concerns, we respectfully request the reviewer to reassess our work and consider increasing the rating. Thank you.

---

> > > > ### Author Response · Authors · 2024-11-26
> > > >
> > > > Dear Reviewer FJwk,
> > > >
> > > > We sincerely request the reviewer to review the rebuttal that we had provided. We believe we have addressed all the comments and experiments raised by the reviewer through our detailed responses and clarifications. We would appreciate your prompt attention before the discussion phase concludes. A thorough reassessment will help to ensure a fair evaluation of our work. Thank you for your immediate attention to this matter.
> > > >
> > > > Best regards, The Authors

---

> > > > > ### Author Response · Authors · 2024-11-28
> > > > >
> > > > > Dear Reviewer FJwk,
> > > > >
> > > > > We would like to respectfully inquire whether there are any remaining concerns or aspects of our paper that are unclear and that we can address during the rebuttal period. As this period is meant to resolve issues raised during the review process, we kindly request that the reviewers provide us with more specific feedback or highlight any areas that may still require improvement, so that we can better address them.
> > > > > If there are no further issues, we would greatly appreciate hearing whether the clarifications and revisions we have provided have sufficiently addressed reviewer FJwk’s concerns, and if reviewer FJwk would be willing to reassess our work.
> > > > > Thank you.
> > > > >
> > > > > Best,
> > > > > The Authors

---

> > > > > > ### Author Response · Authors · 2024-12-02
> > > > > >
> > > > > > Dear Reviewer FJwk,
> > > > > >
> > > > > > We sincerely ask Reviewer FJwk whether there are any remaining concerns. In our rebuttal, we have addressed the issues raised by Reviewer FJwk, including additional evaluations on five more models (Video-SALMONN, Gemini, Qwen-Audio, Video-LLaMA2, and LLaVA-OneVision), four of which were specifically suggested by the reviewer. Since this is the last day of the discussion period, if there are no further issues, we would greatly appreciate it if our work could be reassessed. Thank you.
> > > > > >
> > > > > > Best, The Authors

---

### Official Review · Reviewer_MmyJ · 2024-11-03

**Soundness:** 3
**Presentation:** 4
**Contribution:** 3
**Rating:** 8
**Confidence:** 4

**Summary:**

This article proposes a comprehensive benchmark for evaluating the perceptual and understanding capabilities of audiovisual LLMs, which includes four tasks: Audio-driven Video Hallucination, Video-driven Audio Hallucination, and Audio-visual Matching. It assesses the hallucination phenomena of existing AV-LLMs, as well as the cross-modal matching and reasoning abilities of these models, and provides relevant analyses and conclusions.

**Strengths:**

1. Well-Written and Accessible: The article is well-written, well-motivated, and easy to follow.
2. Comprehensive Benchmark: The proposed benchmark is comprehensive, containing several complementary dimensions and devised tasks.
3. Valuable Takeaways: The takeaways provide valuable analyses, conclusions, and insights.
4. The paper presents three methods to improve the trustworthiness of multimodal large language models (MLLMs).

**Weaknesses:**

1. How do the authors ensure the quality of the dataset? Are there any evaluation measures in place?

2. For the tasks of Audio-driven Video Hallucination and Video-driven Audio Hallucination, how do the authors ensure that the visuals or audio contain objects that are either silent or not present? Most events, objects, and sounds in videos are quite singular, with the sound-producing objects being consistent and uniform in both video and audio.

3. How should the presence of ambient sounds, such as wind or rain, which do not correspond to specific "objects" in the visuals, be handled?

4. If there are multiple objects of the same type in the video that do not make sounds, for example, several dogs that are silent while a dog off-screen is barking, or if the audio consists largely of narration while the video shows people, would such cases still be classified as "present in both video and audio"?

5. How is the situation handled when the audio is background music? Additionally, in cases where the samples themselves are inconsistent between video and audio but originate from the same video source, the model may determine them as not matching, while the ground truth indicates they do match. How is this discrepancy addressed?

**Questions:**

See Weaknesses.

---

> ### Author Response · Authors · 2024-11-22
> **Author response**
>
> We thank reviewer MmyJ for recognizing our work as well-written, well-motivated, and for providing a comprehensive benchmark, valuable takeaways, and methods for improving the trustworthiness of MLLMs, along with constructive feedback. Here, we address the concerns raised by reviewer MmyJ.
>
> **Ensuring the quality of the dataset**
> >**We'd like to emphasize that our dataset is human-verified in all dataset construction stages, which ensures its quality.** Our proposed data generation method effectively transforms the exhaustive human labeling process into a focused verification effort. Thus, our resulting dataset is at the level of humans.
> As mentioned in Section D.4 in the Appendix of our initial submission, we assigned different tasks to each annotator and used majority votes to finalize each QnA label. The first annotator checked the Q&As and audio-visual captions generated with automatic annotations. The second annotator was presented with 10% intentionally inaccurate QnAs (verified by the authors), and successfully revised 9.8% of them. The last annotator annotated everything from scratch. During these phases, 94.4% of the QnA pairs were agreed upon by all three annotators, while 6.6% were agreed upon by two. This process helped to ensure the quality of our dataset.
> Additionally, **we have newly included new error analyses in Section D.5 and Figure S6 of the Appendix in the revision PDF. These sections illustrate how errors are corrected through human verification, further ensuring the dataset's quality.**
>
> **Ensuring the visuals or audio contains objects that are either silent or not present**
> >We recognize that not all videos contain silent objects or out-of-view sound sources (which was also mentioned by Reviewer MmyJ) and already discussed this in Section 3.2 (Lines 232-235) of our initial submission. Consequently, the videos that contain these cases are used for generating QnAs related to hallucinations caused by in-view silent objects or out-of-view sound sources, while others are used to provide QnAs for specific tasks.
> In addition, our Stage 1 audio-visual information disentanglement, followed by human verification, ensures the accuracy of extracting silent visuals and non-present sound sources. As mentioned in lines 239-255 of our initial submission, ChatGPT disentangles in-view and out-of-view information using provided audio captions and visual tags, with human-annotated few-shot examples guiding the process. Furthermore, human verification is conducted to correct any inaccuracies in disentangled information, as detailed in lines 266-271 of the initial submission. If neither in-view silent objects nor out-of-view sound sources exist, they are annotated as "none."
>
> **How can ambient sounds (e.g., wind) be handled?**
> >While not specifically handled, questions that do not make sense (e.g., “Is the wind visible?”) are filtered out by human annotators who checked such queries as “ambiguous” as shown in Figure S5 in the Appendix of the initial submission. However, these ambient sounds are annotated in Stage 1 and can be used to extend our dataset for a single-modal hallucination benchmark, such as Audio-only hallucination.
>
> **Handling multiple objects with the same type but different statuses?**
> >Involving “events” helped to handle multiple objects of the same type with different statuses. For instance, if an in-view person is silent while there is speech happening out-of-view, our Stage 1 pipeline will annotate “person” for “in-view silent object,” and “talking person” for “out-of-view sound source.” Therefore, when generating QnAs in Stage 2, it might be phrased as “Is a talking person visible in the video? No.” Since the person exists in the video but is not talking, this question serves as a hallucination QnA. Furthermore, all annotations in both Stage 1 and Stage 2 are verified by humans, who correct any nonsensical QnAs that were automatically generated by our proposed pipeline.
>
> **Handling inconsistent audio and video signals (e.g., background music)**
> >Although inconsistencies between video and audio from the same source may not affect the construction of cross-modal hallucination pairs, they become critical when managing matching tasks, such as when background music is unrelated to the visual scene. We handle these inconsistencies through human verification. These sources are automatically assumed to be “matched” since they originate from the same video; however, if human annotators consider them unaligned, they adjust the answer accordingly. For instance, in videos with background music, annotators label these as inconsistent and classify them as "not matching," or they label them as "ambiguous" if they are difficult to determine. Finally, we take the majority vote from three annotators to decide the labels for these matchings, thus managing or discarding inconsistent audio-video signals effectively.

---

> > ### Author Response · Authors · 2024-11-25
> >
> > Dear Reviewer MmyJ,
> >
> > We sincerely appreciate reviewer MmyJ’s time and effort in reviewing our submissions and providing valuable feedback. We believe that we have thoroughly addressed all the comments reviewer MmyJ had raised. We welcome any further comments or feedback that may help to enhance our work. We'd be happy to further discuss.
> > Thank you.

---

> > > ### Author Response · Authors · 2024-11-26
> > >
> > > Dear Reviewer MmyJ,
> > >
> > > We sincerely request the reviewer to review the rebuttal that we had provided. We believe we have addressed all the comments and experiments raised by the reviewer through our detailed responses and clarifications. We would appreciate your prompt attention before the discussion phase concludes. A thorough reassessment will help to ensure a fair evaluation of our work. Thank you for your immediate attention to this matter.
> > >
> > > Best regards, The Authors

---

> > > > ### Author Response · Authors · 2024-11-28
> > > >
> > > > Dear Reviewer MmyJ,
> > > >
> > > > We would like to respectfully inquire whether there are any remaining concerns or aspects of our paper that are unclear and that we can address during the rebuttal period. As this period is meant to resolve issues raised during the review process, we kindly request that the reviewers provide us with more specific feedback or highlight any areas that may still require improvement, so that we can better address them.
> > > > If there are no further issues, we would greatly appreciate hearing whether the clarifications and revisions we have provided have sufficiently addressed reviewer MmyJ’s concerns, and if reviewer MmyJ would be willing to reassess our work.
> > > > Thank you.
> > > >
> > > > Best,
> > > > The Authors

---

> > ### Comment · Reviewer_MmyJ · 2024-12-02
> >
> > Thank you for the author's response. All my concerns have been addressed. Overall, I believe this is a very good piece of work with in-depth analysis. Therefore, I will raise my score to 8. Additionally, I have one more question. In the author's response to other reviewers, the relative performance of video-salmonn and video-llama2 varies in the Audio-driven Video Hallucination and Video-driven Audio Hallucination. Could you provide further detailed descriptions of this?

---

> > > ### Author Response · Authors · 2024-12-02
> > >
> > > We sincerely appreciate Reviewer MmyJ for taking the time to review our work and rebuttal, and for reassessing our submission, leading to an increased rating. The feedback provided was valuable in clarifying the scope of our work and enhancing its overall quality!
> > >
> > > **[Additional question]**
> > > >We believe the performance difference, particularly in Video-driven Audio Hallucination, is primarily due to the differences in the training datasets between Video-SALMONN and Video-LLaMA2. Video-LLaMA2 is trained on a diverse range of audio and audio-visual datasets, including WavCaps, AudioCaps, Clotho, VGGSound, UrbanSound8k, ESC50, and others. In contrast, Video-SALMONN is trained on AudioCaps, EGo4D (egocentric videos), and How2 (instructional videos), which may not fully capture the diversity of scenarios associated with general audio events, such as those involving animals, humans, and nature. Since Video-LLaMA2's datasets cover a wider variety of scenarios, they better support learning the complex relationship between audio and video perception, leading to significantly better performance in Video-driven Audio Hallucination.

---

> > > > ### Comment · Reviewer_MmyJ · 2024-12-02
> > > >
> > > > Thank you for your patient response. I have no further questions! I believe this work is of high quality, so I still support its acceptance.

---

> > > > > ### Author Response · Authors · 2024-12-03
> > > > > **Official Comment by Authors**
> > > > >
> > > > > We deeply appreciate your decision to raise the score. Your insightful feedback greatly contributed to enhancing the quality of our paper, and we truly valued the opportunity to engage in meaningful discussions with you.
> > > > >
> > > > > Warm regards,
> > > > > The Authors

---

### Official Review · Reviewer_sQPA · 2024-11-05

**Soundness:** 3
**Presentation:** 3
**Contribution:** 2
**Rating:** 6
**Confidence:** 3

**Summary:**

The paper proposes AVHBench, a benchmark for evaluating cross-modal hallucination for audio-visual language models. The paper finds that the current models fall short when it comes to evaluations designed to test hallucination, with a performance close to random guesses. The eval bench comes out of a GPT-4 aided data generation pipeline with human verification. The author fine-tuned their model using the data coming out of the same pipeline without any evaluation and found that they were able to improve the hallucination issue significantly.

**Strengths:**

Looking into visual-audio-language model cross-modal hallucination seems to be novel. The paper is well-written and clearly motivated.

**Weaknesses:**

* The synthetic dataset seems to be the important component for both the evaluation set and the training set.  There seem to be several sources of error that the author did not either discuss or give some analysis on:
 1. For Audio-Visual disentanglement: (a) error might come from the visual tagging process (b) The prompt in Table s1 can not distinguish the sound or appearance of multiple instances of the same type of object. (e.g. given two people and the sound of a human talking, it will recognize someone is talking but have no idea whether the one who is talking is in view)
2.  For audio-visual caption generation. Given two unaligned audio and visual captions, likewise, the language model is not guaranteed to be able to capture the correspondence between visual and audio information.
Given the above, I think it's crucial that the author give some error analysis of the proposed pipeline (e.g. from the verification data of manual labour.)

* Related to the previous point, the proposed pipeline is not able to generate audio-visual captions/questions that require temporal reasoning. Would be good to have some discussion on this.

* Also related to the above point, to show the proposed pipeline is truly useful for generating data to fine-tune the audio-visual model, it would be nice to see some more results on how the fine-tuned model performs on other audio-visual benchmarks.(In Table 4 beyond VAST Captioning dataset)

**Questions:**

My concern majorly lies in the error analysis and limitation of the data generation pipeline as well as the results on the fine-tuned models.

---

> ### Author Response · Authors · 2024-11-22
> **Author response (1/2)**
>
> We thank reviewer sQPA for recognizing our work as novel, well-written, and clearly motivated, along with providing constructive feedback. We address the concerns raised by sQPA below and encourage the reviewer sQPA to check the newly updated revision PDF that includes the requested content. The red-colored texts indicate the newly added part in the revision PDF.
>
> **Error analysis**
> >We appreciate reviewer sQPA for suggesting us to conduct an error analysis to improve our paper. First of all, **we’d like to highlight that our final data is human-verified data, and initial errors should be corrected by humans.** Our proposed data generation method is actually an efficient and scalable human annotation tool that converts from an exhaustive human labeling process to a focused verification one. Thus, our data is at the level of humans. Nonetheless, **according to the request, we have analyzed and revealed how much human verification plays a crucial role in identifying and correcting errors that occur in automatically generated annotations.** We have newly added this error analysis in Section D.5 and Figure S6 in the Appendix of the revision PDF and provided the details below.
>
> >**(1) (audio-visual disentanglement) Error in the visual tagging**
> >- Errors in visual tagging are handled by Stage 1 human annotators. As we use visual tagging extracted from an off-the-shelf model to disentangle audio-visual information, errors are sometimes included. However, as shown in Figure S6-(a) in the Appendix of the revision PDF, human annotators review all disentangled information and correct any incorrect disentangled visual information. For example, the automatically disentangled in-view silent object "plane," which does not exist in the video but is captured in the visual tagging, is successfully removed by the annotators.
>
> > **(2) (audio-visual disentanglement) Distinguishing multiple instances of the same type of object**
> >- Our work primarily serves as a corner research that first evaluates and reveals cross-modal hallucinations, i.e., assessing how one modality might affect the perception of another. We mainly focus on scenarios where out-of-view sound sources affect visual perception in audio-visual LLMs, or in-view silent objects affect audio perception. Thus, our evaluation does not include the fine-grained analysis that the reviewer mentioned, such as distinguishing which of two visible people made a sound. However, we agree that incorporating this finer evaluation into future work could make our benchmark more comprehensive. We have newly added Section 5 Limitation and future work in the revision PDF, and included this discussion in Line 475-479.
>
> >**(3) (audio-visual caption generation) Capturing correspondence between unaligned visual and audio captions**
> >- Human verification is also important in handling inaccuracies in the initial audio-visual captions due to unaligned visual and audio captions. During this rebuttal period, our analysis revealed that 33% of captions were corrected by human annotators. Although our automatic pipeline effectively predicted 67% of the captions accurately, human verification proved essential for ensuring accuracy. An example highlighting the necessity of this verification can be seen in the newly added Figure S6-(b) in the Appendix of the revision PDF. Here, the given audio caption describes, "A woman speaking as water softly splashes and trickles while wind lightly blows into a microphone," while the video caption incorrectly states, "A man sitting on a large rock in the middle of a lake," as the video shows a woman sitting on the rock. The initial audio-visual caption generated by ChatGPT, therefore, inaccurately merges these descriptions, resulting in: "A woman speaks softly as water splashes and trickles with a man sitting on a large rock in the middle of a lake." However, human verification identifies this error, and corrects it from “with a man” to “and she is,” ensuring the refined audio-visual caption accurately describes the video: “A woman speaks softly as water splashes and trickles, and she is sitting on a large rock in the middle of a lake.”
>
> >Thanks for the suggestion, which improves the completeness of our work.

---

> > ### Author Response · Authors · 2024-11-22
> > **Author response (2/2)**
> >
> > **Adding discussion about audio-visual temporal reasoning**
> > >Thanks for highlighting this important point. As videos increase in length and complexity, we agree that incorporating temporal reasoning could indeed enhance the dataset's comprehensiveness. We believe that extending the dataset with source video datasets that include temporal QnAs [C1, C2] could facilitate constructing audio-visual captions with temporal reasoning. We have newly added Section 5 Limitation and future work of the revision PDF, and included this discussion in Line 475-481.
> >
> > >[C1] Yi et al., Learning to Answer Questions in Dynamic Audio-Visual Scenarios, CVPR22
> >
> > >[C2] Yang et al., AVQA: A Dataset for Audio-Visual Question Answering on Videos, MM22
> >
> > **Evaluation on other audio-visual benchmark**
> > >As requested by reviewer sQPA, we have conducted additional evaluations on an audio-visual benchmark to further demonstrate the effectiveness of enhancing robustness against cross-modal hallucinations in zero-shot scenarios.
> > Specifically, we evaluate the fine-tuned models on AVinstruct, an audio-visual joint instruction benchmark that includes both open-ended (audio-visual captioning) and closed-ended tasks (multiple-choice questions). We observed consistent improvements across both tasks, indicating the effectiveness of our proposed pipeline for generating data to fine-tune the AV-LLMs. We have newly included these results in Table 5 in Section 4 of the revision PDF.
> > |              |              | VAST      |           |             | AVInstruct |           |             |            |          |
> > |--------------|--------------|-----------|-----------|-------------|------------|-----------|-------------|------------|----------|
> > | **Aligning** | **Finetune** | METEOR(↑) | CIDEr (↑) | GAVIE-A (↑) | METEOR (↑) | CIDEr (↑) | ROUGE-L (↑) | BLEU-4 (↑) | ACC. (%) |
> > | -            | -            | 18.2      | 0.2       | 4.04        | 45.9       | 14.5      | 35.3        | 12.8       | 43.6     |
> > | &#10004;     | -            | 19.2      | 20.7      | 3.68        | 42.2       | 27.1      | 41.5        | 14.9       | 52.6     |
> > | -            | &#10004;     | 18.7      | 13.4      | 2.58        | 53.5       | 76.4      | 52.3        | 25.1       | 44.2     |
> > | &#10004;     | &#10004;     | **22.1**      | **47.6**  | **5.09**    | **58.1**   | **102.0**   | **55.8**    | **28.5**    | **57.8**  |

---

> ### Author Response · Authors · 2024-11-25
>
> Dear Reviewer sQPA,
>
> We sincerely appreciate reviewer sQPA’s time and effort in reviewing our submissions and providing valuable feedback. We believe all the comments and experiments suggested by the reviewer sQPA have been addressed, including the error analyses and additional evaluation on another benchmarks. We welcome any further comments or feedback that may help to enhance our work. We'd be happy to further discuss.
> Thank you.

---

> > ### Author Response · Authors · 2024-11-26
> >
> > Dear Reviewer sQPA,
> >
> > We sincerely request the reviewer to review the rebuttal that we had provided. We believe we have addressed all the comments and experiments raised by the reviewer through our detailed responses and clarifications. We would appreciate your prompt attention before the discussion phase concludes. A thorough reassessment will help to ensure a fair evaluation of our work. Thank you for your immediate attention to this matter.
> >
> > Best regards, The Authors

---

> > > ### Author Response · Authors · 2024-11-28
> > >
> > > Dear Reviewer sQPA,
> > >
> > > We would like to respectfully inquire whether there are any remaining concerns or aspects of our paper that are unclear and that we can address during the rebuttal period. As this period is meant to resolve issues raised during the review process, we kindly request that the reviewers provide us with more specific feedback or highlight any areas that may still require improvement, so that we can better address them.
> > > If there are no further issues, we would greatly appreciate hearing whether the clarifications and revisions we have provided have sufficiently addressed reviewer sQPA’s concerns, and if the reviewer sQPA would be willing to reassess our work.
> > > Thank you.
> > >
> > > Best,
> > > The Authors

---

### Author Response · Authors · 2024-11-22

**We thank the reviewers for their constructive comments and suggestions for improving our work. We appreciate the positive feedback:**
- Well-written, clearly motivated, and well-designed figures (All the reviewers)
- Providing novel cross-modal hallucination task (Reviewer sQPA)
- Comprehensive benchmark containing several complementary dimensions and devised tasks (Reviewer MmyJ and 7vLX)
- Valuable analyses, insights, benchmarks, and takeaways (Reviewer MmyJ and FJwk)
- Presenting methods to improve the trustworthiness of MLLMs (Reviewer MmyJ)
- Reasonable semi-automatic pipeline (Reviewer FJwk)
- Providing new perspectives on the study of multimodal hallucinations (Reviewer huna)

**We have addressed all the comments from the reviewers, and we look forward to additional feedback or constructive discussion for clarification. Here is the summary of the rebuttal:**
- We conducted the evaluations of our benchmark with recent audio-visual models including VideoLLaMA2, Video-SALMONN, and Gemini (Flash), and uni-modal models, including LLAVA-OneVision and Qwen2-Audio, and newly added these analyses to Appendix Section C  and Table S1 in the revision PDF.
- We conducted an error analysis on our automatically constructed dataset with human verification and newly updated these in Appendix Section D.5 and Figure S6 in the revision PDF.
- We evaluated our fine-tuned models on an additional audio-visual benchmark to demonstrate the effectiveness of our fine-tuning approach, with updates in Table 5 of the revision PDF.
- We have newly added Section 5 "Limitations and Future Work" in the revision PDF to more thoroughly present the limitations of our work and discuss future directions for enhancing dataset comprehensiveness.
- We have newly added Appendix Section H, "Maintenance" in the revision PDF, where we outline our plan to maintain a benchmark leaderboard.
- We provided clarifications and extended explanations in response to each reviewer’s comments.

**We would like to remind our key contributions:**
- Proposing AVHBench, the first benchmark designed to assess cross-modal hallucination in audio-visual LLMs
- Proposing a semi-automatic annotation pipeline that reduces manual labeling costs while ensuring high-quality annotations
- Analyzing the presence of cross-modal hallucination, and investigating causes of such hallucination using proposed AVHBench
- Demonstrating insights for enhancing robustness of audio-visual LLMs against cross-modal hallucination.

Given these solid contributions, we would like to highlight that, as acknowledged by all the reviewers, our research is notable as an important cornerstone research, despite room for improvement. All the comments are well-taken, and we believe that our submission has further improved through this revision. Thanks.

---

### Meta-Review · Area_Chair_Ghbk · 2024-12-22

**Metareview:**

Paper proposes an Audio-Visual LLM benchmark. The paper was reviewed by five expert reviewers and received the following ratings: 4 x marginally above the acceptance threshold and 1 x accept, good paper. Reviewers agree that the paper is well-written and clearly motivated. At the same time, a number of concerns were brought up by reviewers, centering around: (1) potential issues (e.g., errors) with data stemming from the synthetic dataset [sQPA, MmyJ], (2) lacking evaluations with more SoTA models [FJwk, huna] and (3) lacking diversity in the dataset [7vLX] (e.g., in terms of scenes). Authors have provided an extensive rebuttal that addresses these concerns and reviewers generally found compelling. Additional SoTA evaluations were also provided. Unfortunately, not all reviewers engaged in post-rebuttal discussion, but it appears that corresponding comments have been adequately addressed.

AC has read the reviews, rebuttal and corresponding discussion as well as looked at the paper itself. While AC is somewhat concerned that there is no discussion of how proposed benchmark differs from others (e.g., AVSBench, AVQA, AV-Odyssey), the task of AV understanding is reasonably novel with fewer benchmarks generally available. As such, AC believes that proposed AVHBench would likely be valuable for driving the research forward in the community and is recommending Acceptance.

**Additional Comments On Reviewer Discussion:**

Authors have provided an extensive rebuttal that reviewers mostly found convincing. Specifically, reviewer [sQPA] mentions that he/she "appreciates the author's clarification on error and additional experiments on other audio-visual benchmarks" and retains the score; reviewer [Mmyj] acknowledges that "all my concerns have been addressed" and reviewer [huna] thanks authors "for the detailed response" while acknowledging that he/she is still concerned with "visual hallucinations caused by the speech content". Unfortunately, [FJwk] and
[7vLX] did not engage in discussion or responded to the rebuttal despite multiple prompts from the authors and AC. Overall, the discussion converged to paper being rather borderline-ish, however, after careful consideration and discussion with Senior AC, the recommendation above has emerged.

---

### Decision · Program_Chairs · 2025-01-22

Accept (Poster)